

# Comparison and validation of eight satellite rainfall products over the rugged topography of Tekeze-Atbara Basin at different spatial and temporal scales

Tesfay G. Gebremicael[1, 2, 3], Yasir A. Mohamed[1, 2, 4], Pieter van der Zaag[1, 2]Amdom G. Berhe[5], Gebremedhin G. Haile[3], Eyasu Y. Hagos[5], Mulubrhan K. Hagos[3]

[1]IHE Delft Institute for Water Education, P.O. Box 3015, 2601 DA Delft, The Netherlands
[2]Delft University of Technology, P.O. Box 5048, 2600 GA Delft, The Netherlands
[3]Tigray Agricultural Research Institute, P.O. Box 492, Mekelle, Ethiopia
[4]Hydraulic Research Center, P.O. Box 318, Wad Medani, Sudan
[5]Mekelle University, P.O. Box 231, Mekelle, Ethiopia

*Correspondence to:*T.G. Gebremicael(t.gebremicael@un-ihe.org/dutg2006@gmail.com)

## Abstract

Satellite rainfall products are considered important options for acquiring rainfall estimates in the absence of ground measurements. However, estimates from these products need to be validated as their accuracy can be affected by geographical position, topography, and climate, as well as by the algorithms used to derive rainfall from satellite measurements. Eight satellite-based rainfallproducts (TRMM, CHIRPS, RFEv2, ARC2, PERSIANN, GPCP, CMAP and CMORPH) were evaluated against ground observations over the complex topography of the upper Tekeze-Atbara basin in Ethiopia. The performance was evaluated at various temporal (daily, monthly, seasonal) and spatial (point, sub-basin, basin) scales over the period 2002-2015.
Results show thatCHIRPS, TRMM,and RFEv2 performed well and wereable to capture the rainfall measured by rain gauges. The BIASand correlation of these products were within $\pm 25\%$ and $>0.5$over different time steps.The remaining products poorly performed at daily time step with higher BIAS (up to $\pm 200\%$) and lower correlation ($<0.5$). CMORPH, PERSIANN, and ARCv2 were relatively better while CMAP and GPCP performed poorly ($r<0.4$) in all conditions. The overall performance of all products was lower in the mountainous areas of the basin with station elevation$>2500$ m.a.s.l. Compared to the lowlands, the BIAS at highlands increased by 35% whilst the correlation dropped by 28%. Underestimation and overestimation of rainfall dominated in the mountainous and lowland areas, respectively.CMORPH and TRMM overestimated while the remaining products underestimated the rainfall at all spatiotemporal scales. CMAP, ARC2, and GPCP estimates were the most affected by large underestimation. Unlike in temporal scale, the performance of the products did not show a uniform pattern with respect to spatial scale.Their performance improved from point to aerial comparisons in the lowlandswhereas it slightly reduced athighland areas. Poor performance in the highlands contributed to a slightly lower performance at basin scale compared to thepixel-to-pointcomparison.
Our results showthat rainfall estimates from CHIRPS and TRMM have a consistently good agreement with ground rainfall at different spatiotemporal scalesin the upper Tekeze-Atbara basin. Interpolating the sparse and unevenly distributed rain gauges over the complex terrains however introduces unknown uncertainties with respect to the actual rainfall.

**Keywords:**Rainfall estimation**,**Satellite products,validation, Nile River Basin, Ethiopia



# 1. Introduction

2 Accurate information on rainfall data is necessary for many operational and research fields of water management, hydrological applications, and agricultural forecasts (Guo & Liu, 2016;

4 Sunilkumar *et al.*, 2015). It is arguably considered as the most important driving force for any hydrological model. Despite its importance for socioeconomic development, ground-based

6 rainfall measurements are sparse and unevenly distributed,especially in developing countries(Behrangi *et al.*, 2011; Gebremichael *et al*., 2014). The recommended density of

8 ground rainfall measuring network in tropical regions is one gauge per 600 - 900 km$^2$for flat and 100 - 250km$^2$ for mountainous areas, respectively (WMO, 1994). However, such

10 densities are not available in most tropical regions (Taye & Willems, 2013; Worqlul *et al.*, 2014). Due to different limiting factors, including climatic conditions and human geography,

12 ground rainfall stations are sparse or do not exist at the required temporal and spatial scales (Meng *et al.*, 2014). Recently, satellite rainfall products are considered as important

14 alternative options for acquiring rainfall estimates. These products are advantageous in terms of temporal and spatial coverage and providing data sources in ungauged basins (Dinku *et al.*,

16 2014; Katsanos *et al.*, 2016).

18 Satellite rainfall products are increasingly available with almost global coverage and the supply of those products are becoming cost effective sources for hydrological applications

20 (Meng*et al*., 2014; Thiemig et al., 2012). The spatiotemporal resolutions and measurement accuracy of these products are continuously improving because of advancement in sensor

22 technologies and estimation techniques. A number of higher resolution rainfall products are now available at a quasi-global scale (Behrangi *et al.*, 2011; Jiang *et al.*, 2012). The Tropical

24 Rainfall Measurement Mission (TRMM), African Rainfall Estimation (RFE), African Rainfall Climatology (ARC), Global Precipitation Climatology Project (GPCP), Precipitation

Estimation from Remotely Sensed Information using Artificial Neural Networks (PERSIANN)Climate Hazards Group InfraRed Precipitation with Stations (CHIRPS) and

CPC Morphic technique (CMORPH) are among the common products that have been widely applied.

However, satellite rainfall products need to be validated as their accuracy can be affected by

32 geographical position, topography, and climate, as well as by the algorithms used to derive rainfall from satellite measurements(Meng *et al.*, 2014; Xue *et al.*, 2013). Several studies on

34 the validation and comparisons of these products with ground measurements have been





conducted at different scales (e.g. Dinku *et al.*, 2007; Feidas, 2010; Guo & Liu, 2016; Hessels, 2015; Hu *et al.*, 2014; Jiang *et al.*, 2012; Thiemig *et al.*, 2012; Worqlul *et al.*, 2014). Nevertheless, the performance varies among the rainfall products because of different data sources and retrieving algorithms (Derin & Yilmaz, 2014; Toté *et al.*, 2015). In addition, the performance also varies for the same data type across different regions and seasons (Gebremichael *et al.*, 2014; Hu *et al.*, 2014). This indicates that the performance of satellite products largely depends on the location, topography, season, and hydro-climatic characteristics of the study area. Therefore, the reliability of satellite rainfall needs to be validated and compared against ground measurements to a specific area and temporal scales before it can be used in any subsequent application (Feidas, 2010; Ouma *et al.*, 2012).

Validation and inter-comparison of different rainfall products over the complex topography of the upper Tekeze-Atbara (T-A) basin are essential to determine which product is representative. A number of studies have been conducted in Ethiopia to evaluate different satellite rainfall products (e.g., Dinku *et al.*, 2007; Gebremichael *et al.*, 2014; Haile *et al.*, 2013; Worqlul  *et al.*, 2014; Beyissa *et al*., 2017). However, these studies have mainly focused on the Upper Blue Nile basin and to some extent on central Ethiopia. In the T-A basin, where these products can contribute to better understanding of catchment response to land degradation and environmental rehabilitation programs, there has been no comprehensive validation studies. Therefore, this studywas intended to validate eight of the widely used satellite rainfall products on different spatiotemporal scales. The relationships between satellite rainfall products and topography were also carefully explored in order to understand possible errors produced by the rugged terrains.

## 2. Study area and Data

### 2.1. Study area

This study was conducted in the Upper T-A basin, one of the main tributaries to the Nile river located in the Northern Ethiopia, with a total catchment area of 45,694 km$^2$ at the outlet.  It is situated between $37.5^o – 39.8^o$ E and $11.5^o – 14.3^o$ N(Fig.1). The basin is characterized by rugged topography with a significant variation ranging from 833 to 4530 m.a.s.l. About 0.2%, 52%, 42% and 0.62% of the land is found below 1000, between 1000-2000, 2000-3000, 3000-4000 and above 4000 m.a.s.l, respectively. This clearly indicates that topography is a key factor in influencing  microclimates in the basin.



Figure 1: Location map and distribution of rainfall stations in the Upper T-A Basin

The basin is characterized by a semi-arid climate in the east and north and partly semi-humid in the south (Belete, 2007). More than 85 % of the total annual rainfall falls in the wet season (June -September) which varies from 400 mm yr$^{-1}$ in the east to more than 1200 mm yr$^{-1}$ in the south (Fig. 2a). The variations are mainly associated with the seasonal migration of the inter-tropical convergence zone (ITCZ).The beginning and ending of the ITCZ over highlands of Ethiopia varies annually, which mostly causes the inter-annual rainfall variability (Selshi&Zanke, 2004;Nyssen *et al.*, 2005).

The general pattern of rainfall over the basin isalso modified by the complex topography (Dinku *et al.*, 2007; Viste & Sorteberg, 2013).This implies that the movement ofair moisture is substantially modified to create contrasting rainfall regimes in the region (Huber *et al*., 2006). The sudden changes in elevation can obstruct the air mass movement to create a microclimate at the bottom of mountains or can updraft over the mountains to create orographic rainfall (Dinku *et al.,* 2007).

In most regions, rainfall increases with elevation due to the orographic uplifts (Moreno *et al.,* 2014;worqlul *et al*., 2014). However, this relationship is not uniform in the T-A Basin (Fig.2b). Rainfall in the mountains is higher in some areas and lower in others (Kiros*et al*., 2015). Figure 2a indicates that the total annual rainfall increases with elevation in the southern and southwestern parts of the basin only. In contrast, it reduceswith elevation in most other parts of the basin.Stations located in the highlands of the eastern and northern parts of the basin receiveless rainfall compared to the associated lowlands (Fig. 2a). This is attributed to the complex local topography, which alters proximity to the sources of moist air and seasonal movementsof the ITCZ (Van der Ent*et al*., 2010; Kiros *et al*., 2016).

During the rainy season, the ITCZ moves towards the Northern part of the basin, which brings moisture from the Atlantic and Indian oceans through westerly (Degefu *et al*., 2016; Mohamed *et al*., 2005). When the rain-bearing winds reach the basin, their direction is modified by the local topography forcing the release of moisture in the lower areas before they reach the top of mountains. This creates more intense and shorter duration convective rainfall events in the lowlands where warm and moist airflows encounter the mountain foothill. Van der Ent *et al*. (2010)showed that topography can play an important role in





moisture cycling either by blocking or capturing moving air masses. Another possible reason for the low rainfall over the northern-eastern highlands is that whereas here the eastern rain-bearing winds are stronger, they carry less water vapour (Viste & Sorteberg, 2013). The non-uniform patterns of rainfall against the topography can strongly influence the performance of satellite rainfall estimates (Haile *et al.*, 2013).

Figure 2: Relationship between rainfall and elevation in the T-A basin

## 2.2. Datasets

### 2.2.1. Rain gauge data

Ground rainfall data used for validation of the satellite products comprised of 34 stations located within and surrounding the basin (Fig.1). These data were provided by the Ethiopian Meteorological Service Agency (NMA). The datasets coverdaily data for the period from 2002 to 2015. Although the number of stations is relatively good, their distribution over the basin is not uniform. Most of the gauges are located in easily accessible areas and the distribution of gauges in the lowland areas are sparse (Fig. 1). Interestingly, most of the rainfall stations with a relatively good quality of data are located in the highland areas where the spatial variability of rainfall is very high.A summary of these ground measurements with vertical locationsis given as an electronic supplementary file (Table S1).

Quality control of rainfall data from each station was done to identify if there were outliers and missing values. All outliers were then compared to neighbouring gauges to cross-check if observed extreme values resulted from extreme climate events. Stations with large data gaps in between the selected validation period were excluded from the analysis. After data screening, 34 stations out of the 75 in the basin were found to be reliable with a relatively consistent record.

### 2.2.2. Satellite rainfall products

The validation and inter-comparison of eight satellite rainfall products were performed atdaily, monthly, and seasonal scales. Table 1 provides the summary of satellite rainfall products used for this study. These products were selected based on their public domain and long-term data available, spatiotemporal resolution, near-real-time availability and their common applications in Africa (Dembélé & Zwart, 2016; Dinku *et al.*, 2007; Thiemig *et al.*, 2012).



Table 1: Summary of selected satellite rainfall products for this study (in descending order of spatial resolution)

The CHIRPS datasets, developed by the US Geological Survey (USGS) and the Climate Hazards Group at the University of California are blended products which combine global climatologies, satellite observations and in-situ rainfall observations from Global Telecomunications system (GTS) (Funk *et al.*, 2014; Knapp *et al.*, 2011). CHIRPS incorporates $0.05^o$ resolution satellite rainfall estimates with in-situ station data to produce daily time series (Katsanos *et al.*, 2016).

ARCv2 is produced by the National Oceanographic and Atmospheric Administration Climate Prediction Center (NOAA-CPC) and provides daily rainfall data over Africa. It is very similar to RFEv2 except the 30 minutes is replaced by the 3-hourly IR data (Love *et al.*, 2004).

The RFEv2 is also provided by NOAA-CPC for Famine Early Warning Systems Network to assist in disaster-monitoring activities over Africa (Herman *et al.*, 1997). RFEv2 has been operational since 2001 and uses rainfall estimates from PM sensors, IR data from METEOSAT and daily rainfall from the GTS reports. Daily rainfall estimates were obtained at $0.1^o$ spatial resolution by merging these sources.

The CMORPH product that produces global rainfall analysis at a very high spatial and temporal resolution is also a product from NOAA-CPC. Unlike the other products, the CMORPH product is not an algorithm for merging of the PM and IR estimates rather it uses the IR information for the spatial and temporal evolution of clouds, not the rainfall estimates (Asadullah *et al.*, 2008; Joyce *et al.*, 2004). It uses rainfall estimates derived from low orbit PM observations and propagate these features using a high temporal and spatial resolution IR data (Joyce *et al.*, 2004). According to Dinku *et al.* (2007), the CMORPH combines the superior retrieval accuracy of the PM and higher resolution of IR data. This method is highly flexible as it allows incorporation of any rainfall estimate from PM satellites.

The PERSIANN precipitation estimates were developed by the Center for Hydrometeorology and remote sensing atthe University of California (Ashouri *et al.*, 2015). It uses an artificial neural network approach to merging the IR and PM data and the rainfall estimates are based on the infrared brightness temperature image provided by geostationary satellites (Hsu *et al.*,

1997). The rainfall estimates in PERSIANN algorithm are available at $0.25^{o}$ spatial resolution.

The latest version of TRMM product (3B42V7) was developed by the National Aeronautics and Space Administration (NASA). This product was obtained from the TRMM Multi-satellite precipitation analysis (TMPA) algorithm which combines Infrared (IR) and Passive Microwave (PM) data retrievals (Guo & Liu, 2016; Huffman *et al.*, 2007). TRMM rainfall estimates incorporates gauge data for bias correction from several sources including national and regional meteorological services. (Funk *et al*., 2014). The TRMM3B43 rainfall products were agreggated from the TRMM3B42 3-hourly estimates and merged with station data to produce daily rainfall (Dinku *et al.*, 2007).

The GPCP is a blended product which combines the Global Precipitation Climatology Center (GPCC) gauge data with the PM and IR rainfall estimates (Huffman *et al.*, 1997). The PM estimates in this product are based on the Special Sensor Microwave /Imager (SSM/I) data from the Defence Meteorological Satellite program (DMSP, US) while the IR data came mainly from Geostationary Operational Environmental Satellite (GOES) Precipitation Index (PI) (Xie & Arkin, 1995). This technique is advantageous as it combines rainfall estimate information from many data sources by taking the strength of each data type.

CMAP products include monthly and pentad (5-day) mean rainfall estimates at $2.5^{o}$ spatial resolution (Feidas, 2010). These techniques produce rainfall estimates by merging ground station data with rainfall estimates from several satellite-based algorithms (Xie & Arkin, 1997). As described in Xie and Arkin (1997), inputs are derived by combining of geostationary and polar orbiting infrared, PM retrievals and rain gauge observations. First, the IR and PM rain estimates are merged using a maximum likelihood approach where the estimate with weights are derived by comparison to the gauge analysis. Then, the gauge analysis is used to obtain an absolute value of the merged product (Feidas, 2010).

## 3. Methodology

### 3.1. Validation processes

The spatial patterns of eight satellite products were evaluated and compared with rain gauge data at daily, monthly, and seasonal scales. Both the satellite and gauge rainfall data were





collected at different temporal scale and first, the daily data were aggregated to monthly and
seasonal scales. As more than 85% of the total annual rainfall occurs during the wet season
(June-September) (Gebremicael *et al.*, 2017), seasonal comparison was considered only for
this period. The ability to replicate the observed rainfall by the products was done during this
common period between all satellite and station rainfall. Considering the given climatic
variability, complex topographical characteristics and hydrological working units of the basin,
the performance of these products were evaluated using two approaches, namely point-to-
pixel and aerial averaged rainfall comparison.

Rainfall over a complex topography like the T-A basin is largely subjected to small-scale
variability, which implies that evaluation of such satellite products should be at the smallest
possible spatial and temporal scales (Thiemig *et al.*, 2012). Accordingly, in the first approach
all satellite rainfall products from the corresponding grid cell were compared to the ground
observed data within the satellite box. The variance of satellite estimate is smoother in space
and time as these products are represented by the spatial averages over the pixels. For this
analysis, the satellite rainfall products were extracted for the location of each rainfall station
and their performance were evaluated using statistical indices. It was assumed that the amount
of point rainfall is uniform in the area of the pixel which may not necessarily true.The second
approach was based on the aerial rainfall comparison at different spatial scales.
Representative sub-basins from lowland and highland areas (Fig.1) with an average elevation
of 1400 and 3000 m.a.s.l.were considered in order to account for the effect of topography.
Satellite products were validated at sub-basin and basin level by comparing spatially
aggregated pixel values against a corresponding interpolated observed rainfall from gauge
stationsusing the inverse distance weighting (IDW) method (Ruelland *et al.,* 2008).

### 3.2. Evaluation statistics

The satellite rainfall products were quantitatively evaluated against ground observations using
four statistical indices:the relative percent of bias (PBIAS), Pearson correlation coefficient (r),
Root Mean Square Error (RMSE), and Mean Absolute Error (MAE) Table 2). A detailed
description of these indices can be found in Toté *et al.* (2015) and Thiemig *et al.* (2012).

Table 2: Statistical indices used for the satellite rainfall products performance evaluation

Where xi is observed rainfall from raingauge, yi is satellite rainfall product, N is the number of pairs of products,
$\bar{x}$ and $\bar{y}$are the average of observed and satellite rainfall data, respectively.





2    Agreement between estimates and observation is considered satisfactory for PBIAS and r

values ±25% and > 0.5, respectively (Moriasi *et al.*, 2007). The lower the RMSE and MAE

4    values, the closer the satellite estimates are to the ground measurements. The unit of RMSE

and MAE is mm/time period.

## 6    4. Results and discussion

### 4.1. Comparison at pixel-to-point spatial scale

8    The performance of satellite estimates was evaluated by comparing these data for 34 rainfall

stations at grid level covering the location of the station. Comparisons were carried out at

10    daily, monthly, and seasonal periods. First, dailyrainfall of eight products was compared with

the observed daily rainfall. Figure 3 shows the PBIAS (%) and correlation (r)of all satellite

12    estimates against ground station values. The daily estimates performed poorly in the majority

of stations. However, CHIRPS, RFEv2 and TRMM had a relatively good performance with

14    lower PBIAS, RMSE and MAE and higher r compared to other products (Fig.3 andTables S2-

and S3). The average value of PBIAS for all stations were -13%, -16% and 17% for CHIRPS,

16    RFEv2 and TRMM, repectively. Similarly, r value of these products was≥ 0.5 in the majority

of stations with an average value of  0.52, 0.50 and 0.50, respectively.. The RMSE and

18    MAE,which evaluates the average magnitude error between satellite estimates and ground

stations showed the same trend as PBIAS and r (Tables S2 and S3).The remaining products

20    failed to capture the observeddaily rainfall with correlationof < 0.5 and higher PBIAS, RMSE

and MAE in most stations (Tables S2 and S3). The ARC2, GPCP and CMAP performed

22    poorly.

24    The precision of these products to reproduce the observed rainfall wasfurther investigatedat

monthly time series. Table 3 shows the average value of  accuracy indicators obtained by

26    comparing each product with ground stations.The results indicate thatthe performance of all

products improvedwhen daily data are aggregated to monthly data. The correlationfor

28    CHIRPS, RFEv2 and TRMM were>0.5 in all stations with an average value of 0.61, 0.59 and

0.56, respectively (Table3). Similarly, the PBIAS value reducedat monthly time scale.The

RMSE and MAE indices also decreased at monthly scale, which implies the agreement

between satellite and ground rainfall increased (Table 3).Forexample, Fig. 4 compares the

pattern of statistical indices for all products in four representative (highland, lowland and



medium) stations. CHIRPS, RFEv2 and TRMM outperformed to the other products. ARCv2,

2 CMAP and GPCP again performed poorly with a r < 0.5 and higher PBIAS (Table 3).

4 Figure 3: Comparison of daily satellite rainfall estimate with ground measurements, (a) PBIAS, (b) Correlation (r)

Table 3: Average accuracy indicators obtained from monthly comparison

Moreover, comparisons based on average monthly point rainfall (2002-2015) at the given

10 locations indicate that rainfall estimates of CHIRPS, RFEv2 and TRMM products agree with the corresponding ground measurements (Fig.5). Monthly rainfall patterns from these

12 products have a consistent and strong agreement with the ground rainfall compared to the remaining products.

To gain further information on the seasonal variations of rainfall estimate skills of the satellite

16 products, comparisons were also madefor the entire rainy season (June-September). Figure 6 presents an inter-comparison of wet season rainfall estimates with the observed rainfall of the

18 same period. The spatial distribution of correlationcoefficients (Fig. 6) and PBIAS (Fig.S1) show patterns that are similar to these ofthe daily and monthly results. However, the

20 performance of all products was significantly improved during the wet season. Six satellite products had an excellent agreement with ground rainfall (CHIRPS, TRMM, PERSIANN,

22 RFEv2, ARC2, and CMORPH) during the wet season.CHIRPS, RFEv2, and TRMM correlated best with the observed rainfall compared to the remaining products. With average

24 values of 0.84, 0.74, and 0.75 for CHIRPS, RFEv2, and TRMM, respectively, the correlation coefficient of these products showed a strong agreement. The PBIAS of these productswas

26 also within the range of ±25%in most stations. The RMSE and MAE indices were also lower than the other products (Tables S4 and S5). Next, the PERSIANN, CMORPH and ARCv2

28 products showed a good agreement with the gauged rainfall. Improved correlation (r>0.5)and lower PBIAS, RMSE and MAE were obtained in 80 % of the stations, however the GPCP and

30 CMAP products continued to show poor agreement despite some improvements (Fig. 6 and Tables S4 and S5).

32

Figure 4: Monthly statistical indices at pixel to point rainfall comparison.





2          Fig 5: Comparison of mean monthly rainfall (2002-20015) at four representative ground stations

Point-to-pixel comparison of the different temporal scales showed that all satellite products

6   suffer from both over and underestimations, explained by negative and positive values of PBIAS.Both phenomena were observed in all products at several locations and time

8   scales.TRMM and CMORPH systematicaly overestimate the rainfall in more than 20 stations while they underestimate rainfall in the remaining stations. The RFEv2, GPCP, ARC2 and

10   CMAP products consistently underestimate rainfall in the majority of ground stations. Most overestimations(underestimations) were observed during the dry (October-May) (wet (June-

12   September)) months (Fig.5). However, the performance of all products to capture the observed rainfall were better in the dry months. This is due to the reduced probability of

14   rainfall during the dry months.

16   Figure 6: Spatial distribution of correlationcoefficients (r) during the wet season comparison.

18   It is also important to remark that inconsistent estimation of rainfall by all products is likely due to the effect of rugged terrains.The overall performance of the satellite rainfall products is

20   lower in the peripheries of the basin where most stations are located in the mountainous area with an elevation> 2,500 m.a.s.l. (Fig.7). As shown in Fig. 4, lower correlation and higher

22   PBIAS is observed in Debark compared to Sekota with an elevation of 3,000 and 1,960 m.a.s.l., respectively.A relatively better performance occurred in central, eastern, and north-

24   western parts of the basin where stations are located below 2,500 m.a.s.l. As an example, the long-term annual observed and satellite rainfall from CHIRPS were plotted against elevation

26   of stations (Fig.7). The graph clearly shows that the relationship pattern of rainfall with elevation is not straightforward.The correlation of these products showed a poor agreement at

28   higher elevation.A similar study in the neighbouring Upper Blue Nile basin by Gebremichael*et al*.(2014) also showed that satellite products failed to capture the ground

30   rainfall in mountainous compared to lowland areas.This result is consistent with other studies carried out elsewhere(e.g. Asadullah *et al.*, 2008; Derin & Yilmaz, 2014; Guo & Liu, 2016;

32   Hu *et al*., 2014)





Figure 7: Relation between annual average rainfall (gauged and CHIRPS)and
elevation

In summary, the combination of daily, monthly, and seasonal point comparisons demonstrate
that the CHIRPS, RFEv2, TRMM, and PERSIANN products have the best agreement with the
observed rainfall across the basin. The evaluation indices at different time scales and average
monthly plots comparison show that CHIRPS performs best, followed by RFEv2 and TRMM.
Over- and under-estimation of daily, monthly and seasonal rainfall by CHIRPS was smaller
compared to the other products.

### 4.2. Comparison based on aerial averaged rainfall

Spatiotemporally aggregated aerial rainfall of each product was also compared with the
corresponding interpolated rainfall from the gaugesat daily, monthlyand seasonal time scales.
Table 4 shows the performance of all products at basin level anddifferent time scale. Similar
to the point-to-pixel comparison, the CHIRPS, RFEv2, TRMM, PERSIANN, and CMORPH
aerial rainfall estimates had the best accuracy,with PBIAS within ±25% at all temporal scales.
However, all products showed a lower performance in terms of correlation with <0.5 and
higher RMSE and MAEat daily compared to monthly and seasonal time scales. Lower
performance at daily scale can be explained by erroneous (non-detection) of more localized
convective rainfall events.The CMAP,ARCv2, and GPCP continued to show lower r and
higher RMSE, MAE and PBIAS consistently atall time scales. The performance improved at
monthly and seasonal scales (Table4). Higher accuracies at larger time scales are due to the
fact that the errors at smaller time scale are symmetrical and offset each other when
aggregated.

The negative and positive values of PBIAS in Table 4 confirms that most products
underestimated rainfall during the wet season, except TRMM and CMORPH. Figure 8 shows
the visual comparison of the long-term monthly aerial average of satellite estimates with the
corresponding ground aerial rainfall. It clearly indicates that the TRMM and CMORPH
products consistently overestimated, whereas CMAP, GPCP and ARC2 underestimated
rainfall in all months. The remaining products showed a varied picture for the different
months. For example, CHIRPS and RFEv2 slightly underestimatedrainfall during the rainy
months of July and August whilst overestimatingrainfall in the dry months (Fig.8).Duringthe
wet season CHIRPS satellite estimatesoutperformed the other products.





Table 4: Comparison of satellite and observed aerial rainfall, at basin scale, at different time scales

Figure 8: Monthly average satellite estimates and ground rainfall comparison at basin scale

4 Whereas all satellite rainfall estimates showed a consistent improving pattern with increasing time scale, their performance did not show a uniform pattern with increasing spatial

6 scale.Figure9 shows a compares average correlation of the products at different spatial scales forthe wet season. Most products performed worse at basin scale compared to pixel-to-point

8 and lowland sub-basin scales. The likely reason is that the aerial averaged rainfall over the complex topography suffers from limitations due to the uneven distribution of rain gauges.

10 The performance of CHIRPS, TRMM, and CMAP improved at basin level compared to pixel-to-point scale whilst all other products performed worse(Fig.9). The relatively poor

12 performance at the basin scale for most products is likely due to the topographical variations across the basin. Variations of topography can significantly compromise the interpolation of

14 observed rainfall (Thiemig *et al.*, 2012). The rainfall stations are also sparsely and unevenly distributed over the basin, which can be a source of systematic errors when interpolating

16 aerial rainfall (Dembélé & Zwart, 2016; Toté *et al.*, 2015).

18 To further understand the effect of complex terrains on the performance of the satellite products, the seasonal aerial rainfall of representative highland and lowland sub-basins was

20 compared (Fig.9). The result ofthe two contrasting topographic featuresdemonstrates that theoverall correlation of satellite rainfall estimates is better in lowland than in high

22 mountainous areas. This suggeststhat the satellite products may not accurately capture the spatial pattern of seasonal rainfall in complex topographic areas like the T-A basin. Further,

24 all products overestimate rainfall in the lowlands and underestimate rainfall of the rain in the highlands.However, comparing both topographic features, the magnitude of underestimation

26 was greater than that of overestimation in most products. For example, CHIRPS, TRMM, and CMORPH underestimated the wet season rainfall over the highland area by 32, 28, and 52%,

28 while it overestimated by 18, 21, and 28% the lowland rainfall, respectively.

30 Fig.9: Comparison of seasonal averaged correlation at a pixel, sub-basin, and basin scales

32 In summary, based on the comprehensive evaluation at different temporal and spatial scales, the CHIRPS, RFEv2, and TRMM outperformed the other satellite rainfall products at all



spatiotemporal scales. The better performance of CHIRPScan be explained by the fact that it

2     to considers topographic effects and its high spatial resolution (Katsanos *et al.*, 2016).The

good performance ofTRMM,and RFEv2 is possibly due to the fact that these productshave a

4     bias correction that is based rain gauge data (Thiemig *et al.*, 2012).

6     Our findings are in agreement withsimilar studies (e.g. Dembélé & Zwart, 2016; Hessels,

2015; Katsanos *et al.*, 2016; Bayissa *et al*., 2017; Dinku et al., 2008). Hessels (2015)

8     compared 10 satellite products over the Nile basin and CHIRPS and TRMM werefound to be

the best-performing products.Bayissa*et al.* (2017)revealed that CHIRPS estimates showed

10     better performance than PERSSIAN and TARCAT over the Upper Blue Nile basin. Similarly,

Gebremichael*et al.* (2014) and Dinku*et al.* (2007)showedthat CMORPH and TRMM well

12     performed in the rugged terrains of neighbouring basin.Next tothese products, CMORPH,

PERSIANN, and ARCv2 were better in capturing the observed rainfall while CMAP and

14     GPCP poorly performed at all spatiotemporal scales.Dinku et al. (2007) also showed that

CMAP and GPCP products poorly performed compared to TRMM and CMORPH in

16     Ethiopia.

18     The performance of all products consistently correlated best with ground measurements when

aggregated at larger time scales. Improved performance with increasing time step is obviously

20     due to counterbalancing of variabilities when accumulated from smaller to larger time scales.

Many studies (e.g. Dembele&Zewart, 2016; Guo& Liu, 2016; Meng*et al.,* 2014)reported that

22     the performance of satellite estimates improved as time step increased. In contrast, the

performance of these products wasnot uniform with an increasing spatial scale. The

24     performance of all products increased from point rainfall to aerial rainfall in the lowlands,

whereas their performance decreased in thehighlands (Fig.9). Apoorer performance in the

26     mountainous area is notable for all products. This result is in agreement with other studies

(e.g. Derin and Yilmaz; 2014; Dinku *et al*., 2007)whichindicates that satellite rainfall products

28     have challenges to estimate orographic precipitation in basins with a complex topography.

## 5. Conclusions and Recommendations

30     This study evaluated the performance of eight-satellite based rainfall products ranging from

high to low resolution over the T-A basin. These products were evaluated and compared

32     withground stations during 2002-2015. A comprehensive approach was applied that

includedpoint-to-pixel and aerial averaged comparisons at different spatialand temporal scales



(daily, monthly, and seasonal). The relationship between rainfall and elevation wasalso analysed to identify the effects of topography on the performance.

The results showed that the CHIRPS, RFEv2, and TRMM rainfall estimates outperformed the other products consistently across all temporaland spatial scales. These products achieved acceptable correlation coefficients (>0.5) and PBIAS, RMSE and MAEvalues for both approaches and at all time scales. The PBIAS of these products were within ±25% atall timescales. CMORPH, PERSIANN, and ARC2 achieved lower scores.  The performance of CMAP and GPCP was poor over the various conditions with PBIAS ranging from -250% to 118% and correlation <0.5. A relatively lower performance is notable for all products in the mountainous areas.

The agreement between the products and rain gauge improved with increase in time scale. This is due to the fact that errors at smaller time scales offset each other when aggregated. All satellite estimates suffered from under- and over-estimation during the different time and spatial scales. Underestimation dominated in the mountainous areas. TRMM and CMORPH overestimated rainfall whilst the remaining products underestimated rainfall consistently at all spatiotemporal scales.CMAP, ARC2, and GPCP estimates were the most affected by large underestimations across all stations. Another key finding of this study isthat unlike in time, the performance of the products did not show a uniform pattern at different spatial scales. The performance improved when increasing the aerial averaged rainfall in the lowlands, whereas it decreased at larger spatial scale in the highlands.Accuracy indicators at point-to-pixel comparison wereslightly better than aerial averaged rainfallin the whole basin.Poor performance over the mountain areas contributed to lower performance at larger spatial scales. Moreover, systematic errors during the interpolation of observed rainfall over the complex topography of the basin might havecontributed to the overall lower performance at the basin scale.

The ranking of these products may not be absolute as validation of these products in different study periods could result in different rankings. Interpolation of the sparse and unevenly distributed rain gauges over complex terrainsmay also introduce significant uncertainties and therefore limits the validity of the result. However, considering the current data availability, the result of this study provides a basis for the utilization of satellite rainfall estimates over the complex topography of the T-A basin. It will be a good reference for future applications of satellite rainfall, especially in rain gauge sparse and ungauged basins with rugged terrains.





2    **Acknowledgement:** This study was carried out with the support of the Netherland Fellowship

Programme (NUFFIC) and the Tigray Agricultural Research Institute (TARI). The authors

4    would like to thank the Ethiopian National Meteorological Agency for providing the weather

data.

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

32

34

36

38

40

42

44

46

48

50

52





Table 1: Summary of selected satellite rainfall products for this study (in descending order of spatial resolution)

| Product | Temporal resolution | Spatial resolution | Coverage | Starting date |
|---|---|---|---|---|
| CHIRPSv8 | Daily | $0.05^{o}$ | $50^{o}$N-$50^{o}$S,$0^{o}$-$360^{o}$E | 1981 |
| ARCv2 | Daily | $0.1^{o}$ | $40^{o}$N-$40^{o}$S,$20^{o}$W-$55^{o}$E | 1983 |
| RFEv2 | Daily | $0.1^{o}$ | $40^{o}$N-$40^{o}$S,$20^{o}$W-$55^{o}$E | 2001 |
| CMORPH | 3 hourly | $0.25^{o}$ | Global | 2002 |
| PERSIANN | Daily | $0.25^{o}$ | Global | 1983 |
| TRMM 3B42v7 | Daily | $0.25^{o}$ | $50^{o}$N-$50^{o}$S,$0^{o}$-$360^{o}$E | 1998 |
| GPCP | Daily | $1.0^{o}$ | Global | 1979 |
| CMAP | pentad | $2.5^{o}$ | Global | 1998 |

Table 2: Statistical indices used for the satellite rainfall products performance evaluation



| Statistical measure | Equation | Ranges | Perfect score |
|---|---|---|---|
| Root Mean Square Error (RMSE) | $\sqrt{1/N \sum_{i=1}^{n}(yi\text{-}xi)^2}$ | 0 to ∞ | 0 |
| Mean Absolute Error (MAE) | $1/N\sum|yi - xi|$ | 0 to ∞ | 0 |
| Percent of bias (PBIAS) | $((\sum yi\text{-}\sum xi)/\sum xi)*100$ | 0 to ∞ | 0 |
| Pearson correlation coefficient (r) | $\dfrac{\sum(xi - \bar{x})(yi - \bar{y})}{\sqrt{(xi - \bar{x})^2}\sqrt{(yi - \bar{y})^2}}$ | -1 to 1 | 1 |

Where xi is observed rainfall from rain gauge, yi is satellite rainfall product, N is the number of pairs of products, $\bar{x}$ and $\bar{y}$ are the average of observed and satellite rainfall data, respectively.



Table 3: Average accuracy indicators obtained from monthly comparison

| Satellite estimate | PBIAS (%) | r | RMSE (mm/month) | MAE (mm/month) |
|---|---|---|---|---|
| CHIRPS | -8 | 0.61 | 17 | 16 |
| ARCv2 | -50 | 0.43 | 112 | 98 |
| RFEv2 | -10 | 0.56 | 23 | 21 |
| CMORPH | 14 | 0.48 | 75 | 29 |
| PERSIANN | -11 | 0.52 | 41 | 23 |
| TRMM | 7 | 0.59 | 32 | 26 |
| GPCP | -27 | 0.36 | 143 | 103 |
| CMAP | -42 | 0.32 | 126 | 83 |



2    Table 4: Comparison of satellite and observed aerial rainfalls at basin level with different time scale

| Indices | Temporal scale | CHIRPS | ARC2 | RFEv2 | CMORPH | PERSIANN | TRMM | GPCP | CMAP |
|---|---|---|---|---|---|---|---|---|---|
| PBIAS | Daily | *-10* | -58 | *-8* | *18* | *-13* | *11* | -41 | -38 |
| | Monthly | *-8* | -41 | *-6* | *15* | *-11* | *8* | -28 | -33 |
| | Wet season | *-6* | *-21* | *-3* | *11* | *-8* | *6* | *-19* | *-24* |
| r | Daily | 0.49 | 0.22 | 0.48 | 0.19 | 0.28 | 0.41 | 0.15 | 0.21 |
| | Monthly | *0.69* | 0.39 | *0.5* | 0.41 | *0.54* | *0.56* | 0.3 | 0.28 |
| | Wet season | *0.88* | 0.41 | *0.72* | *0.55* | *0.64* | *0.7* | 0.36 | 0.39 |
| RMSE (mm) | Daily | 36 | 111 | 50 | 42 | 48 | 25 | 89 | 60 |
| | Monthly | 42 | 178 | 61 | 70 | 49 | 34 | 205 | 80 |
| | Wet season | 79 | 201 | 67 | 112 | 137 | 142 | 395 | 309 |
| MAE (mm) | Daily | 19 | 102 | 25 | 39 | 48 | 25 | 79 | 54 |
| | Monthly | 41 | 141 | 36 | 68 | 51 | 31 | 131 | 78 |
| | Wet season | 71 | 198 | 68 | 102 | 131 | 123 | 171 | 229 |

Values in italics indicate acceptable region of PBIAS (-25% - 25%) and correlation coefficient (r>0.5)




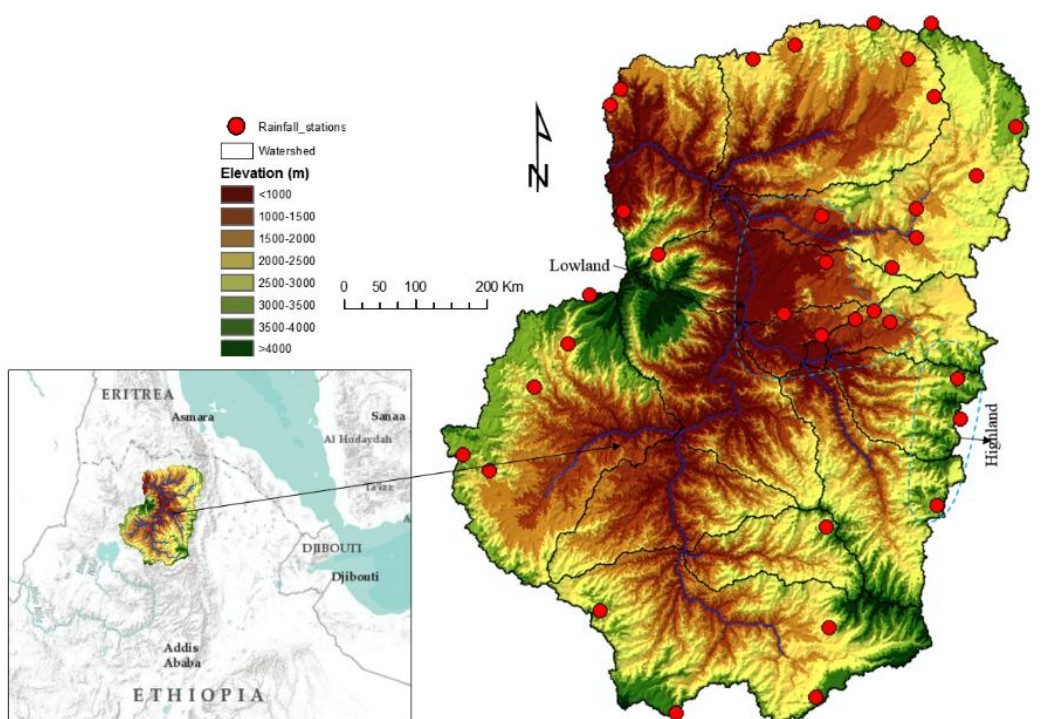

Figure 1: Location map and distribution of rainfall stations in the Upper T-A Basin





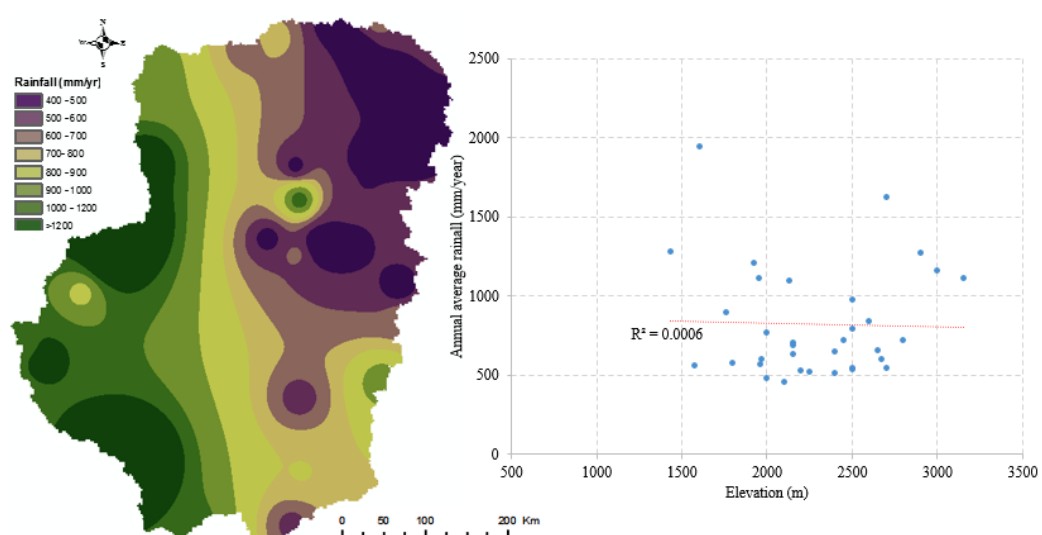

Figure 2: Relationship between rainfall and elevation in the T-A basin





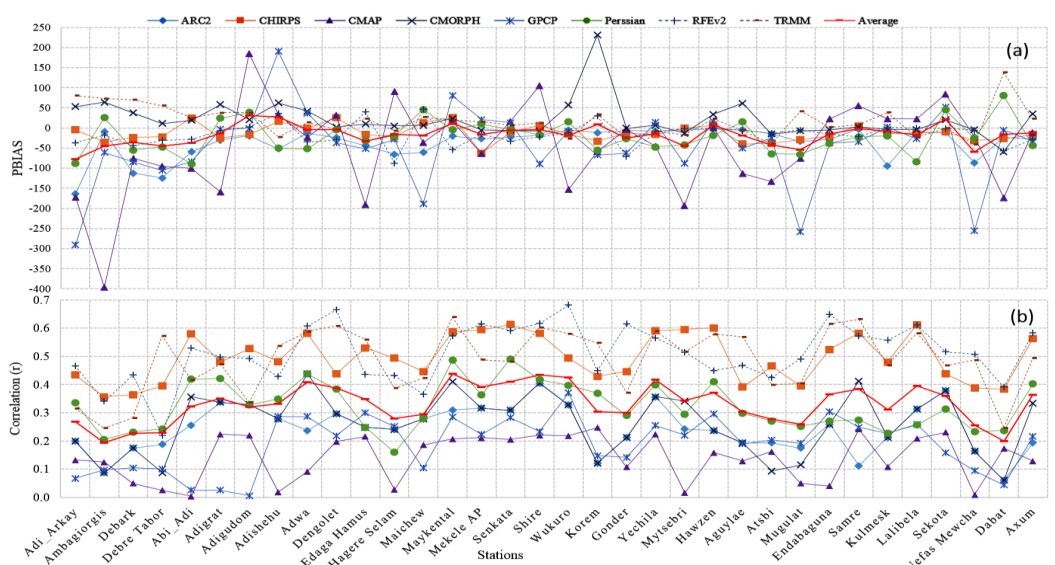

12    Figure 3: Comparison of daily Satellite rainfall estimate with ground measurements, (a)
PBIAS, (b) Correlation (r)



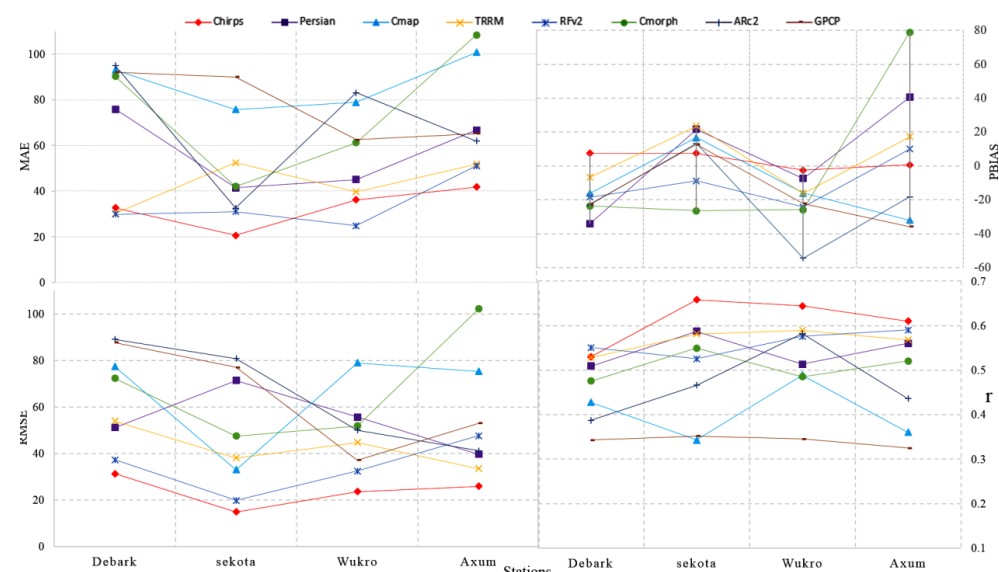

Figure 4: Monthly statistical indices at pixel to point rainfall comparison.



14      Fig 5: Comparison of mean monthly rainfall (2002-20015) at four representative
        ground station



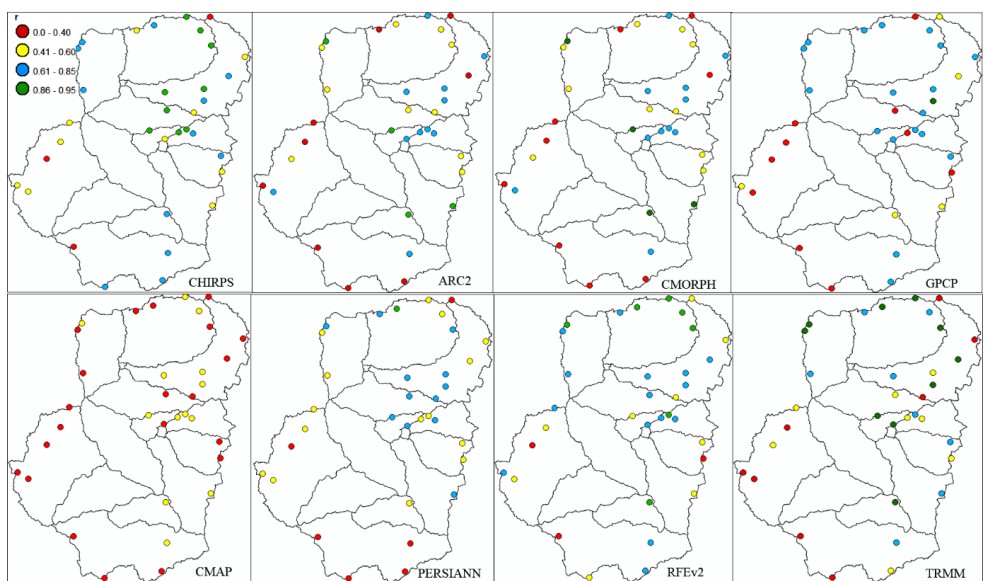

2    Figure 6: Spatial distribution of correlationcoefficients (r) during the wet season comparison.





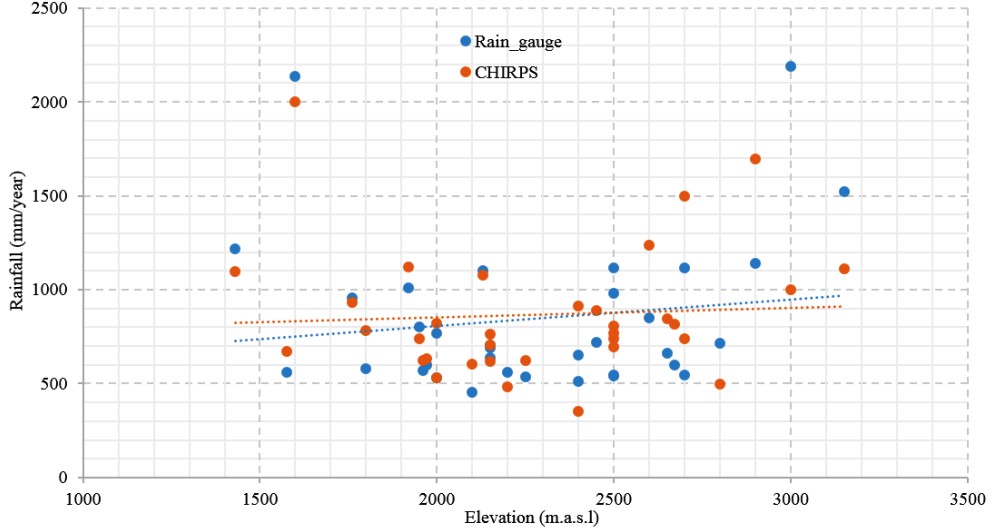

Figure 7: Comparison of annual average rainfall with elevation





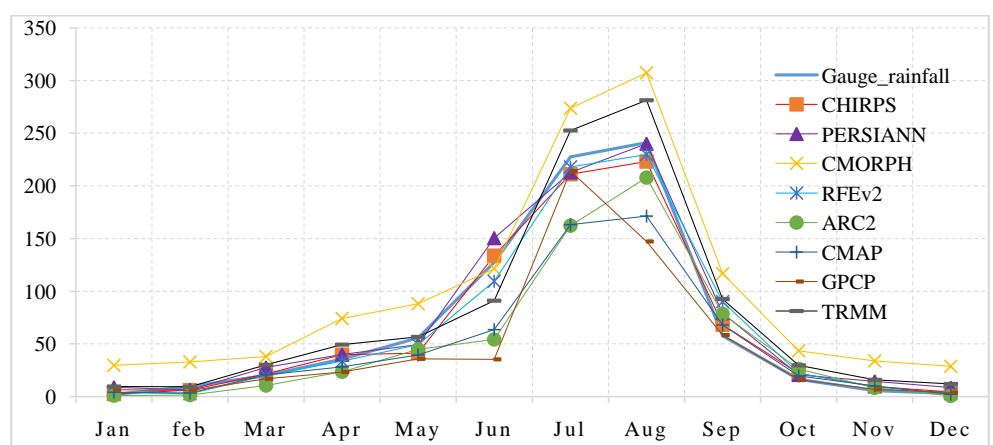

6     Figure 8: Monthly average satellite estimates and ground rainfall comparison at basin scale



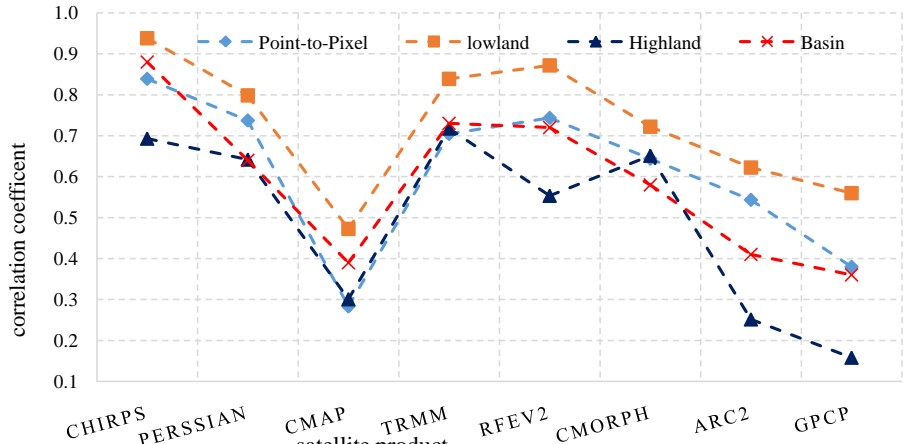

Fig.9: Comparison of seasonal averaged correlation at a pixel, sub-basin, and basin scales

