# Peer review of "Comparison and validation of eight satellite rainfall products over the rugged topography of Tekeze-Atbara Basin at different spatial and temporal scales"

_Hydrology and Earth System Sciences, 2017_

## Referee Comment (RC1) · Anonymous Referee #1 · 10 Oct 2017

Authors conducted a comprehensive evaluation of eight remote sensing rainfall products over T-A basin. It is an important step before applying remote sensing rainfall in hydrologic and/or agricultural application. However, the quality of this manuscript should be further improved to meet the criteria of HESS. My main concerns are listed as:

1) The independent of gauged rainfall to satellite rainfall. As we know, authors are also mentioned that satellite products like TRMM are calibrated by gauged rainfall at monthly scale. Therefore, authors are required to identify whether the 34 stations

were used by satellite rainfall products or not. If used, what kind of impact should be anticipated?

2) Some similar and new references are missed. Recently, several papers discussed the topography impacts on the satellite rainfall in mountainous regions, such as Tibet (Xu, 2017) and Mekong(He, 2017; Wang 2017), as:

–Xu R., F. Tian, L. Yang, H. Hu, H. Lu, and A. Hou (2017), Ground validation of GPM IMERG and TRMM 3B42V7 rainfall products over southern Tibetan Plateau based on a high-density rain gauge network, J. Geophys. Res. Atmos., 122, doi:10.1002/2016JD025418.

–He, Zhihua, Long Yang, Fuqiang Tian, Guangheng Ni, Aizhong Hou, Hui Lu. Inter-comparisons of Rainfall Estimates from TRMM and GPM Multisatellite Products over the Upper Mekong River Basin. JOURNAL OF HYDROMETEOROLOGY, 18:413-430.

–He, ZhihuaïijŇHongchang Hu, Fuqiang Tian, Guangheng Ni and Qingfang Hu. Cor-recting the TRMM rainfall product for hydrological modelling in sparsely-gauged moun-tainous basins. Hydrological Sciences Journal, 2017 VOL. 62, NO. 2, 306–318.

–Wang, Wei, Hui Lu, Tianjie Zhao, Lingmei Jiang, Jianchen Shi, Evaluation and Com-parison of Latest GPM and TRMM Products over Mekong River Basin, IEEE JOUR-NAL OF SELECTED TOPICS IN APPLIED EARTH OBSERVATIONS AND REMOTE SENSING, 10(6), 2540-2549, DOI:10.1109/JSTARS.2017.2672786, 2017

3) GPM is not used in this study, but it should be introduced in the introduction part, as it is the state-of-art satellite rainfall product.

4) Information of eight products. It is recommended to include more details information of these eight products, since it would help to explain the different performances of them.

5) In evaluation statistics: it is recommended to use relative RMSE, and please use RRMSE to evaluate the performance of the eight products. A Taylor diagram may be a

good choice for comprehensive evaluation.

6) P11, L12-14, if authors want to compare the performance in wet months with that in dry months, please show the PBIAS, RRMSE,R of different period. Then, we can quantitatively evaluate the performance.

7) More discussion about why chirps outperforms others is needed! Why CMAP and GPCP are worst? Related to resolution?

Minor comments 8) P6,L9-11, please confirm which version of TRMM is used. If it is 3B42V7, it is not necessary to introduce 3B43 here.

9) P6,L11-19, it is better to change the order of these two paragraphs

10) P6, L17-18, please define the abbreviation of PM, IR, METEOSAT

11) P11,L31, similar findings from Xu et al (2017) in Tibet.

–Xu R., F. Tian, L. Yang, H. Hu, H. Lu, and A. Hou (2017), Ground validation of GPM IMERG and TRMM 3B42V7 rainfall products over southern Tibetan Plateau based on a high-density rain gauge network, J. Geophys. Res. Atmos., 122, doi:10.1002/2016JD025418.

12) P13,L24, remove "of the rain"

---

## Referee Comment (RC2) · Anonymous Referee #2 · 12 Oct 2017

Review on "Comparison and validation of eight satellite rainfall products over the rugged topography of Tekeze-Atbara Basin at different spatial and temporal scales"

**Manuscript ID: hess-2017-504**

**Authors: Gebremicael et al**

**General comments:**

The manuscript focuses on evaluating of different satellite rainfall products in the Tekeze-Atbara Basn, Ethiopia. It is interesting to see a validation study of satellite data in the hydrologically remote part of the world where there is limited data for understanding the climate and hydrology. However the structure of the paper is not easy and clear and the results are not clearly discussed. There are also many type errors. I see many spacing errors between words. On the abstract section alone I have seen more than 10 errors. I have indicated those errors as minor comments. Those errors are too many to list them in my review; I hope the authors will spend some time to correct those errors. With this and other concerns I have indicated below I cannot recommend to accept the paper for publication.

**Specific comments:**

**Abstract**

- The authors indicated that they have evaluated the performance of the products at various spatial and temporal scale. However, in the abstract Line 26 to 28, the spatial and temporal scale of the evaluation is not indicated.
- 2. The abstract should be rewritten to summarize the evaluation result at the multiple temporal and spatial scale. The authors indicated that they have done evaluated the 8 products at various temporal (daily, monthly, seasonal) and spatial (point, sub-basin, basin) scales. However, they did not indicated clearly which products worked at what scale.

**Introduction**

- 3. Page 2: The statement from 23 to 28 needs a reference.
- 4. The authors indicated that tomography as a key factor influencing microclimates in the basin (Page 3 line 32 -33). However, Figure 2b which indicate the relationship between elevation and annual average rainfall doesn't capture the effect of topography. That relationship between rainfall and topography as indicated in this figure is insignificant. What are the authors claming that the topography is a key factor?
- 5. Figure 2b disproves the stamen on page 4 line 18 and 19.
- 6. The rainfall products were not described very well. As the authors indicated satellite rainfall products quality can be affected by the algorithms used. The authors should discuss the different algorithms and platforms used by those products. What part of the electromagnetic spectrum was used? Are they polar orbiting, or sun synchronized satellites or a geostationary satellites are used? The description of the the different products on page 6 and 7 should address this.
- 7. Page 6 line 21: the autors describe CMORPH product as having a very high spatial and temporal resolution however in the summary table (Table 1) this product doesn't prove to be at a higher resolution compared to others such as CHIRP and ARC and others?
- 8. Page 7 line 4. TRMM 3B42V7 is not a latest version
- Table 1 should indicate that the temporal reolution for TRMM3B42V7 should be 3hr. And the product TRMM3B42 should be referred as TMPA-3B42 (Huffman et al., 2010; Prakash et al., 2013; Vrieling et al., 2010).
- 10. Page 8 line 22 to 24: Why the inverse distance interpolation is selected? And what was the grid size used for interpolation this will matter since your rainfall products have a various spatial resolution? Inverse distance weighting (IDW) is a possible simple way to go but probably not the best one. There are interpolation algorithms that take into account secondary information (e.g. kriging with external drift).
- 11.
- The performance indicators for satellite rainfall are too simplistic. The authors should consider a categorical statistics to evaluate the effectiveness of those satellite images. Refer Haile et al. (2010).

Haile, A.T., Rientjes, T., Gieske, A., Gebremichael, M., 2010. Multispectral remote

sensing for rainfall detection and estimation at the source of the Blue Nile River. International Journal of Applied Earth Observation and Geoinformation, 12: S76-S82. The authors should indicate the number of incorrect and correct rain detection by those satellite products.

Why the authors include RMSE and AME isnot RMSE better explanatory than AME since it gives higher weight for larger errors. Otherwise, they provide similar outputs.

13. The reference use on page 9 line 3 Moriasi et al., 2007 is actually for a performance evaluation of simulated flow, sediment and nutrient. My question is that if you accept a PBIAS of  $\pm$  25 and R of 0.5 (which will be 0.25 R-square) as input to your hydrological model; imagine the performance of your model. I really do not agree with the performance evaluation criteria.

**Result**

- 14. The authors provided a single average statistics like average PBIAS, r, RMSE and MAE for different satellite products (Page 9 line 14 and 15). The authors should discuss the range of variability of those statistics and their relation to landscape position.
- 15. The discussion in line 16 page 9 is lamped. The authors should address the range of variation, standard deviation and their relation to landscape postion. Otherwise this doesn't make any sense "Similarly, r value of these products was≥ 0.5 in the majority of stations with an average value of 0.52, 0.50 and 0.50, respectively.."

What does the average line representing in Figure 3 a and b? what does that implies?

- 16. The autores indicated that RMSE and MAE has showed the same trend as PBIAS and r (page 9 line 18 and 19). How is this measured?
- 17. This doesn't make any sense, the study is about comparing of those products with gauged data, but here they averaged the performance statistics. I guess the authors should discuss the range of performance/variability in terms of spatial and temporal scale for each products since this was indicated on the abstract section as a method (page 1 line 23 and 24).

18.

Tables

- 19. Table 2: On Figure 3a I can see a PBIAS value of negative but under Table 2 the autors indicated rage of PBIAS from 0 to infinity. How do you council that?
- 20. Table 2: The authors should remember that R = 1 doesn't mean perfect, it is obvious we have to check the slope and interest of the fitted line. Eg. Y = 5x + 8 has Pearson correlation coefficient (r) of 1 but Y and X are not similar.
- 21. Table 3 is duplicated on Table 4. Remove Table 3

**Figures**

- 22. Figure 1. Label the two figures. What does the dotted line over the DEM represent?
- 23. Figure 2. Label the two figures and describe them independently.
- 24. Figure 3. What does the average line representing?
- 25. Figure 4 where are those representative station located in the watershed?

General comments:

Abstract: the abstract full or problem

- 1. Line 21: space between rainfallproducts
- Line 26 space between thatCHIRPS, Line 26 space between TRMM, and, Line 26 space between wereable
- 3. Line 27 space between BIAS and
- 4. Line 28 space between >0.5 over different
- 5. Line 35 space between respectively.CMORPH
- 6. Line 35 space between scale. Their
- 7. Line 39 space between lowlandswhereas
- 8. Line 40 space between athighland
- 9. Line 41 space between the pixel-to-point comparison
- 10. Line 42 space between showthat

- 11. Line 42 space between scalesin
- 12. Page 3 line 29 and 30 modify it as: with a significant elevation variation
- 13. Page 6 Line 17: sofar PM and IR are not defined. I see later in the paper they are defined.
- 14. Many many errors (dailyrainfall page 9 line 10, (r)of page 9 line 11, andTables page 9 line 14, double fullstops (page 9 line 17), MAE, which line 18,
- 15. Page 9: wasfurther (line 24), investigatedat, that the, correlationfor, reducedat, Forexample
- 16. Page 10: madefor, correlationcoefficients, of the, season.CHIRPS,
- 17. Page 11: many
- 18. Page 12: so many type errors
- 19. Page 14: ofTRMM, productshave, withsimilar, werefound, products.Bayissaet, (2017)revealed, (2007)showedthat CMORPH, etcccc

**Reference**

- Haile, A.T., Rientjes, T., Gieske, A., Gebremichael, M., 2010. Multispectral remote sensing for rainfall detection and estimation at the source of the Blue Nile River. International Journal of Applied Earth Observation and Geoinformation, 12: S76-S82.
- Huffman, G.J., Adler, R.F., Bolvin, D.T., Nelkin, E.J., 2010. The TRMM multi-satellite precipitation analysis (TMPA), Satellite rainfall applications for surface hydrology. Springer, pp. 3-22.
- Prakash, S., Mahesh, C., Gairola, R., 2013. Comparison of TRMM Multi-satellite Precipitation Analysis (TMPA)-3B43 version 6 and 7 products with rain gauge data from ocean buoys. Remote sensing letters, 4(7): 677-685.
- Vrieling, A., Sterk, G., de Jong, S.M., 2010. Satellite-based estimation of rainfall erosivity for Africa. Journal of hydrology, 395(3): 235-241.

---

## Author Comment (AC1) · 20 Nov 2017

**Author response to reviewer's comments**

On behalf of myself and the co-authors, I take the opportunity to thank the anonymous reviewers for his/her constructive comments, questions and editions. We have responded to all questions and comments, as discussed below. Most of the given comments and suggestions by the reviewer were relevant, and accordingly we have updated the manuscript significantly. We feel the quality and readability of the paper have been improved significantly.

**Reviewer #1**

Authors conducted a comprehensive evaluation of eight remote sensing rainfall products over T-A basin. It is an important step before applying remote sensing rainfall in hydrologic and/or agricultural application. However, the quality of this manuscript should be further improved to meet the criteria of HESS. My main concerns are listed as:

Response: The authors would like to thank the anonymous reviewer for his/her essential comments and suggestions. All given comments and suggestions helped us to improve the readability of the paper. Detailed responses to each specific issues and concerns of the reviewer are given as follows.

1. The independent of gauged rainfall to satellite rainfall. As we know, authors are also mentioned that satellite products like TRMM are calibrated by gauged rainfall at monthly scale. Therefore, authors are required to identify whether the 34 station were used by satellite rainfall products or not. If used, what kind of impact should be anticipated?

   Response: We agree with the reviewer that the gauged rain fall from the ground stations used for evaluation should be independent of the gauged rainfall used for calibration of satellite rainfall. The result of the comparisons can be misrepresented if the same ground rainfall data is used in both calibration of the satellite product and performance evaluation. To avoid such problems, all rain gauge networks applied for the comparison are independent of the Global Precipitation Climatology Centre (GPCC) networks used for calibrations of the satellite products. The manuscript is then improved accordingly.

2. Some similar and new references are missed. Recently, several papers discussed the topography impacts on the satellite rainfall in mountainous regions, such as Tibet (Xu, 2017) and Mekong(He, 2017; Wang 2017), as:
   –Xu R., F. Tian, L. Yang, H. Hu, H. Lu, and A. Hou (2017), Ground validation of GPM IMERG and TRMM 3B42V7 rainfall products over southern Tibetan Plateau based on a high-density rain gauge network, J. Geophys. Res. Atmos., 122, doi:10.1002/2016JD025418.
   –He, Zhihua, Long Yang, Fuqiang Tian, Guangheng Ni, Aizhong Hou, Hui Lu. Intercomparisons of Rainfall Estimates from TRMM and GPM Multisatellite Products over the Upper Mekong River Basin. JOURNAL OF HYDROMETEOROLOGY, 18:413-430.

–He, Zhihuaïj NHongchang Hu, Fuqiang Tian, Guangheng Ni and Qingfang Hu. Correcting the TRMM rainfall product for hydrological modelling in sparsely-gauged mountainous basins. Hydrological Sciences Journal, 2017 VOL. 62, NO. 2, 306–318.
–Wang, Wei, Hui Lu, Tianjie Zhao, Lingmei Jiang, Jianchen Shi, Evaluation and Comparison of Latest GPM and TRMM Products over Mekong River Basin, IEEE JOURNAL OF SELECTED TOPICS IN APPLIED EARTH OBSERVATIONS AND REMOTE SENSING, 10(6), 2540-2549, DOI:10.1109/JSTARS.2017.2672786, 2017.

Response: The authors would like to thank the reviewer for his/her updating us on the recent and similar references. These papers were helpful to discuss and improve our manuscript. Such references are now included in the paper.

3. GPM is not used in this study, but it should be introduced in the introduction part, as it is the state-of-art satellite rainfall product.

Response: we agree with the reviewer that the Global Precipitation Measurement (GPM), which was developed based on the success of TRMM and released since 2014 could also a good option over the basin. However, selected satellite products in this study were based on availability of long-term data and their common application in Africa which the GPM lacks to be included in our study. Even though not included in the comparison, we have introduced it in the introduction as per the reviewer's suggestion.

4. Information of eight products. It is recommended to include more details information of these eight products, since it would help to explain the different performances of them.

Response: We agree with the reviewer that including more detailed information on each satellite product would help to understand their performance. To this effect, we have improved the descriptions and content of information for each product in the document. Moreover, several references are also included to refer readers for further detail explanations of these products.

5. In evaluation statistics: it is recommended to use relative RMSE, and please use RRMSE to evaluate the performance of the eight products. A Taylor diagram may be a good choice for comprehensive evaluation.

Response: We agree that the RRMSE is also one of the statistical indices for measuring of disagreement between predicted values against a given reference values which signifies a smaller discrepancy relative to the predicted values. The RMSE also indicates the discrepancy between observed and forecasted values. Smaller value of bothRMSE and RRMSE indicates a better agreement between observed and forecasted. In the same way, both measurements are limited to assessing the errors between observed and predicted, but they cannot tell how accurate the products are. The RMSE and MAE are the two most commonly recommended measures for assessing the predicted accuracy (Chai and Draxler, 2014;Bayissa et al., 2017; Krause et al., 2005). The RMSE has an advantage of showing bigger deviations and helps to provide a complete picture of the errors distribution. Moreover, majority of recent published papers on satellite rainfall validation

(e.g.Gebremicael et al., 2014; Hu et al., 2014; Behringi et al., 2017; Worqlul et al., 2014; Guo et al., 2014, Dembele et al., 2016; Jiang et al., 2012) including papers suggested by the reviewer (He et al., 2017; Xu et al., 2017) have applied RMSE instead of RRMSE to measure the errors. Given its most commonly applied error measurement tool in satellite rainfall validation studies, we feel that replacing RMSE by RRMSE will not have a significant change in the result of the study.

6.  P11, L12-14, if authors want to compare the performance in wet months with that in dry months, please show the PBIAS, RRMSE,R of different period. Then, we can quantitatively evaluate the performance.

Response: We agree with the reviewer that including quantitative explanation will help to elaborate the differences between dry and wet seasons. Quantitative values of these performance indicators are already summarized in Table 4. Moreover, Fig, 3 and Fig.7 also explains how accurate these products are across all months of the year. However, to make it more clear for readers quantitative examples are now included in the discussion text document and the manuscript is improved accordingly.

7.   More discussion about why chirps outperforms others is needed! Why CMAP and GPCP are worst? Related to resolution?

Response: Better performance of CHIRPS is due to its high spatial resolutions and the efforts made to consider topographic effects comparing to the remaining products. Another possible reason could be due to the fact that these products are linked to their embedded bias correction that relies on rain gauge data. Poor performance of CMAP and GPCP is due to the smaller spatial resolution which is at 2.5º compared to other products (less than 1º for all products).Various studies (e.g, Xie and Arkin, 1997; Feidas, 2010, Dinku et al., 2007) also showed that the CMAP and GPCP product suffers from inhomogeneity in addition to its course spatial and temporal resolutions.  Such explanations are also included in the main document.

8.  P6,L9-11, please confirm which version of TRMM is used. If it is 3B42V7, it is not necessary to introduce 3B43 here.

Response: Thank for your suggestion. We have used TRMM (3B42V7) and no need of to mention 3B43 here. This has also modified in the manuscript accordingly.

9.  P6,L11-19, it is better to change the order of these two paragraphs

Response: Thank you for your suggestion and the paragraphs are now shifted each other.

10. P6, L17-18, please define the abbreviation of PM, IR, METEOSAT

Response: These abbreviations are now defined in the first sentences of the document.

11. P11,L31, similar findings from Xu et al (2017) in Tibet. –Xu R., F. Tian, L. Yang, H. Hu, H. Lu, and A. Hou (2017), Ground validation ofGPM IMERG and TRMM 3B42V7 rainfall

products over southern Tibetan Plateaubased on a high-density rain gauge network, J. Geophys. Res. Atmos., 122,doi:10.1002/2016JD025418

Response: This reference is included in the mentioned sentences

12. P13,L24, remove "of the rain"

Response:  removed

**Reference:**

Bayissa, Y., Tadesse, T., Demisse, G., & Shiferaw, A.: Evaluation of Satellite-Based Rainfall Estimates and Application to Monitor Meteorological Drought for the Upper Blue Nile Basin, Ethiopia. *Remote Sens.*, 9(7), 669, 2017.

Behrangi, A., Behnaz, K., Tsou, C., Amir, A., Kuolin , Soroosh, S., & Bacchetta, N.: Hydrologic evaluation of satellite precipitation products over a mid-size basin. J. Hydrol., 397 225-237, 2015.

Dembélé, M., & Zwart, S. J.: Evaluation and comparison of satellite-based rainfall products in Burkina Faso, West Africa. *Int. J. Remote Sens., 37*(17), 3995-4014, 2016.

Dinku, T., Ceccato, P., Grover-Kopec, E., Lemma, M., Connor, S., & Ropelewski, C.: Validation of satellite rainfall products over East Africa's complex topography. Int.J.Remote Sens., 28(7), 1503-1526, 2007

Feidas, H.: Validation of satellite rainfall products over Greece. *Theor. Appl. Climatol., 99,* 193-216, 2010.

Gebremichael, M., Bitew, M. M., Hirpa, F. A., & Tesfay, G. N.: Accuracy of satellite rainfall estimates in the Blue Nile Basin: Lowland plain versus highland mountain. *Water Resour. Re., 50*(11), 8775-8790, 2014.

Guo, R., & Liu, Y.: Evaluation of Satellite Precipitation Products with Rain Gauge Data at Different Scales: Implications for Hydrological Applications. *Water, 8*(7), 28, 2016.

Haile, A.T., Rientjes, T., Gieske, A., Gebremichael, M.: Multispectral remote sensing forrainfall detection and estimation at the source of the Blue Nile River. InternationalJournal of Applied Earth Observation and Geoinformation, 12: S76-S82, 2010

He, Zhihua, Long Yang, Fuqiang Tian, Guangheng Ni, Aizhong Hou, Hui Lu. Intercomparisons of Rainfall Estimates from TRMM and GPM Multisatellite Products overthe Upper Mekong River Basin. J. Hydrometeorol., 18:413-430.

He, Zhihuaïij NHongchang Hu, Fuqiang Tian, Guangheng Ni and Qingfang Hu. Correcting the TRMM rainfall product for hydrological modelling in sparsely-gauged mountainous basins. Hydrol. Sci. J., 2017, 62 (2), 306–318.

Jiang, S., Ren, L., Hong, Y., Yong, B., Yang, X., Yuan, F., & Ma, M.: Comprehensive evaluation of multi-satellite precipitation products with a dense rain gauge network and optimally merging their simulated hydrological flows using the Bayesian model averaging method. J. Hydrol., 452, 213-225, 2012.

Krause, P., Boyle, D.P., Bäse, F., 2005. Comparison of different efficiency criteria for hydrological model assessment. Adv. Geosci. 5, 89-97.

Sapiano, M. R. P. and Arki, P. A.: An Intercomparison and Validation of High-Resolution Satellite PrecipitationEstimates with 3-Hourly Gauge Data, J. hydrometeorol., 10, 149-166, 2009.

Wang, Wei, Hui Lu, Tianjie Zhao, Lingmei Jiang, Jianchen Shi, Evaluation and Comparison of Latest GPM and TRMM Products over Mekong River Basin, ieee journal of selected topics in applied earth observations and remotesensing, 10(6), 2540-2549, 2017.

Worqlul, A, Maathuis, B., Adem, A. A., Demissie, S. S., Langan, S., & Steenhuis, T. S.: Comparison of rainfall estimations by TRMM 3B42, MPEG and CFSR with ground-observed data for the Lake Tana basin in Ethiopia. *Hydrol. Earth Syst. Sci., 18*(12), 4871-4881, 2014.

Xie, P., & Arkin, P.: Global precipitation: a 17-year monthly analysis based on gauge observations, satellite estimates, and numerical model outputs. *B. Amr. Meteor. Soc., 78*, 2539-2558, 1997.

Xu, R., F. Tian, L. Yang, H. Hu, H. Lu, and A. Hou.: Ground validation ofGPM IMERG and TRMM 3B42V7 rainfall products over southern Tibetan Plateaubased on a high-density rain gauge network, J. Geophys. Res. Atmos., 122,2017.

---

## Author Comment (AC2) · 20 Nov 2017

The comment was uploaded in the form of a supplement:
https://www.hydrol-earth-syst-sci-discuss.net/hess-2017-504/hess-2017-504-AC2-supplement.pdf

---

## Author Comment (AC3) · 22 Nov 2017

**Author response to reviewer's comments**

On behalf of myself and the co-authors, I take the opportunity to thank the anonymous reviewers for their constructive comments, questions and suggestions. We have responded to all questions and comments, as discussed below. Most of the given comments and suggestions were relevant, and we have updated the manuscript accordingly. We think the quality and readability of the paper have improved significantly compared to the previous version.

**Reviewer #1**

Authors conducted a comprehensive evaluation of eight remote sensing rainfall products over T-A basin. It is an important step before applying remote sensing rainfall in hydrologic and/or agricultural application. However, the quality of this manuscript should be further improved to meet the criteria of HESS. My main concerns are listed as:

1. The independent of gauged rainfall to satellite rainfall. As we know, authors are also mentioned that satellite products like TRMM are calibrated by gauged rainfall at monthly scale. Therefore, authors are required to identify whether the 34 station were used by satellite rainfall products or not. If used, what kind of impact should be anticipated?

Response: We agree with the reviewer that, to give sensible results of evaluation, the rainfall data from the ground stations should be independent of the ground data used for calibration of satellite products. In fact, all rain gauge networks data used in this study for the comparison are independent of the Global Precipitation Climatology Centre (GPCC) networks used for calibrations of the satellite products. We have amended the text accordingly in Section 2.2.1. "It is to be mentioned that, the selected rain gauge networks used for calibration Climatology Centre (GPCC) networks used for calibration of satellite products."

2. Some similar and new references are missed. Recently, several papers discussed the topography impacts on the satellite rainfall in mountainous regions, such as Tibet (Xu, 2017) and Mekong (He, 2017; Wang 2017), as:

-Xu R., F. Tian, L. Yang, H. Hu, H. Lu, and A. Hou (2017), Ground validation of GPM IMERG and TRMM 3B42V7 rainfall products over southern Tibetan Plateau based on a high-density rain gauge network, J. Geophys. Res. Atmos., 122, doi:10.1002/2016JD025418.

-He, Zhihua, Long Yang, Fuqiang Tian, Guangheng Ni, Aizhong Hou, Hui Lu. Intercomparisons of Rainfall Estimates from TRMM and GPM Multisatellite Products over the Upper Mekong River Basin. JOURNAL OF HYDROMETEOROLOGY, 18:413-430. -He, Zhihuaïij NHongchang Hu, Fuqiang Tian, Guangheng Ni and Qingfang Hu. Correcting the TRMM rainfall product for hydrological modelling in sparsely-gauged mountainous basins. Hydrological Sciences Journal, 2017 VOL. 62, NO. 2, 306–318. -Wang, Wei, Hui Lu, Tianjie Zhao, Lingmei Jiang, Jianchen Shi, Evaluation and Comparison of Latest GPM and TRMM Products over Mekong River Basin, IEEE

**JOURNAL OF SELECTED TOPICS IN APPLIED EARTH OBSERVATIONS AND REMOTE SENSING, 10(6), 2540-2549, DOI:10.1109/JSTARS.2017.2672786, 2017.**

Response: The authors would like to thank the reviewer for his/her valuable suggestions of recent literature on the subject. We have incorporated these references at the respective places within the Manuscript: Xu, 2017; He et al., 2017 and Xu et al., 2017 in Section 1

**3. GPM is not used in this study, but it should be introduced in the introduction part, as it is the state-of-art satellite rainfall product.**

Response: We agree with the reviewer that the Global Precipitation Measurement (GPM), which was released since 2014 could also a good option over the basin. However, selected satellite products in this study were based on availability of long-term data. However, we have introduced it in the introduction part paragraph 2.

4. Information of eight products. It is recommended to include more details information of these eight products, since it would help to explain the different performances of them.

Response: We have improved the descriptions and content of information for each product in the manuscript, as given in Section 2.2.2. Moreover, several references are also included to refer for further readings about these products.

5. In evaluation statistics: it is recommended to use relative RMSE, and please use RRMSE to evaluate the performance of the eight products. A Taylor diagram may be a good choice for comprehensive evaluation.

Response: We agree that the RRMSE is also one of the statistical indices for measuring disagreement between datasets. But we think that RMSE is more commonly used in satellite rainfall data validation studies, than RRMSE. Furthermore, replacing RMSE with RRMSE will not have a significant change in the result of this study.

The RMSE and MAE are the two most commonly used measures for assessing the predicted accuracy (Chai and Draxler, 2014; Bayissa et al., 2017; Krause et al., 2005). The RMSE has an advantage of showing the bigger deviations and provides a complete picture of the errors distribution. Similar relevant studies used RMSE to validate satellite datasets, including, but not limited to: Gebremicael et al., 2014; Hu et al., 2014; Behringi et al., 2017; Worqlul et al., 2014; Guo et al., 2014, Dembele et al., 2016; Jiang et al., 2012; He et al., 2017; Xu et al., 2017.

6. P11, L12-14, if authors want to compare the performance in wet months with that in dry months, please show the PBIAS, RRMSE, R of different period. Then, we can quantitatively evaluate the performance.

Response: Quantitative values of these performance indicators for wet and dry seasons are already summarized in Table 4. Moreover, Fig, 3 and Fig.7 also explain how accurate these products are across the months of the year. However, to make seasonal comparison more

clear, we have added quantitative examples in the discussion, given in Section 4.1 of the manuscript. We have added "For example, CHIRPS and TRMM products perform better in dry season with PBIAS of 10 and 27 compared to -17 and 8 during the wet season.

**7. More discussion about why chirps outperforms others is needed! Why CMAP and GPCP are worst? Related to resolution?**

Response: We think the better performance of CHIRPS is due to its high spatial resolution and consideration of topographic effects. Another possible reason could be due to the fact that this product is linked to its embedded bias correction that relies on rain gauge data. Poor performance of CMAP and GPCP is due to the coarse spatial resolution (2.5°) compared to the other products (less than 1° for all products).

Some studies, e.g., Xie and Arkin (1997), Feidas (2010), Dinku et al. (2007) also showed that CMAP and GPCP products suffer from inhomogeneity in addition to their coarse spatial and temporal resolution. We have added the explanations in the manuscript at Section 4.2.

8. P6,L9-11, please confirm which version of TRMM is used. If it is 3B42V7, it is not necessary to introduce 3B43 here.

Response: We have used TRMM (3B42V7). We have modified the text in the manuscript accordingly, Section 2.2.2 and Table 1.

9. P6,L11-19, it is better to change the order of these two paragraphs

Response: We have changed the order of paragraphs as suggested, Section 2.2.2.

10. P6, L17-18, please define the abbreviation of PM, IR, METEOSAT

Response: Abbreviations are now defined in the first sentences of the document, as Passive Microwave (PM), Infrared (IR)

11. P11,L31, similar findings from Xu et al (2017) in Tibet. –Xu R., F. Tian, L. Yang, H. Hu, H. Lu, and A. Hou (2017), Ground validation of GPM IMERG and TRMM 3B42V7 rainfall products over southern Tibetan Plateau based on a high-density rain gauge network, J. Geophys. Res. Atmos., 122, doi:10.1002/2016JD025418

Response: Thanks for the suggestion, this reference is included in the manuscript, in Section 1.

12. P13,L24, remove "of the rain"

Response: removed

**Reference:**

- Bayissa, Y., Tadesse, T., Demisse, G., & Shiferaw, A.: Evaluation of Satellite-Based Rainfall Estimates and Application to Monitor Meteorological Drought for the Upper Blue Nile Basin, Ethiopia. *Remote Sens.*, 9(7), 669, 2017.
- Behrangi, A., Behnaz, K., Tsou, C., Amir, A., Kuolin , Soroosh, S., & Bacchetta, N.: Hydrologic evaluation of satellite precipitation products over a mid-size basin. J. Hydrol., 397 225-237, 2015.
- Dembélé, M., & Zwart, S. J.: Evaluation and comparison of satellite-based rainfall products in Burkina Faso, West Africa. *Int. J. Remote Sens.*, *37*(17), 3995-4014, 2016.
- Dinku, T., Ceccato, P., Grover-Kopec, E., Lemma, M., Connor, S., & Ropelewski, C.: Validation of satellite rainfall products over East Africa's complex topography. Int.J.Remote Sens., 28(7), 1503-1526, 2007
- Feidas, H.: Validation of satellite rainfall products over Greece. *Theor. Appl. Climatol.*, 99, 193-216, 2010.
- Gebremichael, M., Bitew, M. M., Hirpa, F. A., & Tesfay, G. N.: Accuracy of satellite rainfall estimates in the Blue Nile Basin: Lowland plain versus highland mountain. *Water Resour. Re., 50*(11), 8775-8790, 2014.
- Guo, R., & Liu, Y.: Evaluation of Satellite Precipitation Products with Rain Gauge Data at Different Scales: Implications for Hydrological Applications. *Water*, *8*(7), 28, 2016.
- Haile, A.T., Rientjes, T., Gieske, A., Gebremichael, M.: Multispectral remote sensing for rainfall detection and estimation at the source of the Blue Nile River. International Journal of Applied Earth Observation and Geoinformation, 12: S76-S82, 2010
- He, Z, Long, Y, Fuqiang, T, Guangheng, N, Aizhong, H, Hui L.: Intercomparisons of Rainfall Estimates from TRMM and GPM Multisatellite Products over the Upper Mekong River Basin. J. Hydrometeorol., 18:413-430, 2017.
- He, Z, Hongchang, H, Fuqiang, T, Guangheng N, Qingfang, H.: Correcting the TRMM rainfall product for hydrological modelling in sparsely-gauged mountainous basins. Hydrol. Sci. J., 62 (2), 306–318, 2017.
- Jiang, S., Ren, L., Hong, Y., Yong, B., Yang, X., Yuan, F., & Ma, M.: Comprehensive evaluation of multi-satellite precipitation products with a dense rain gauge network and optimally merging their simulated hydrological flows using the Bayesian model averaging method. J. Hydrol., 452, 213-225, 2012.
- Krause, P., Boyle, D.P., Bäse, F., 2005. Comparison of different efficiency criteria for hydrological model assessment. Adv. Geosci. 5, 89-97.
- Sapiano, M. R. P. and Arki, P. A.: An Intercomparison and Validation of High-Resolution Satellite Precipitation Estimates with 3-Hourly Gauge Data, J. hydrometeorol., 10, 149-166, 2009.
- Wang, Wei, Hui Lu, Tianjie Zhao, Lingmei Jiang, Jianchen Shi, Evaluation and Comparison of Latest GPM and TRMM Products over Mekong River Basin, ieee journal of selected topics in applied earth observations and remote sensing, 10(6), 2540-2549, 2017.
- Worqlul, A, Maathuis, B., Adem, A. A., Demissie, S. S., Langan, S., & Steenhuis, T. S.: Comparison of rainfall estimations by TRMM 3B42, MPEG and CFSR with ground-observed data for the Lake Tana basin in Ethiopia. *Hydrol. Earth Syst. Sci.*, 18(12), 4871-4881, 2014.
- Xie, P., & Arkin, P.: Global precipitation: a 17-year monthly analysis based on gauge observations, satellite estimates, and numerical model outputs. *B. Amr. Meteor. Soc.*, *78*, 2539-2558, 1997.
- Xu, R., F. Tian, L. Yang, H. Hu, H. Lu, and A. Hou.: Ground validation of GPM IMERG and TRMM 3B42V7 rainfall products over southern Tibetan Plateau based on a high-density rain gauge network, J. Geophys. Res. Atmos., 122, 2017.

---

## Author Comment (AC4) · 22 Nov 2017

**Author response to reviewer's comments**

On behalf of myself and the co-authors, I take the opportunity to thank the anonymous reviewers for their constructive comments, questions and suggestions. We have responded to all questions and comments, as discussed below. Most of the given comments and suggestions were relevant, and we have updated the manuscript accordingly. We think the quality and readability of the paper have improved significantly compared to the previous version.

**Reviewer #2**

**General comment**

The manuscript focuses on evaluating of different satellite rainfall products in the Tekeze-Atbara Basin, Ethiopia. It is interesting to see a validation study of satellite data in the hydrologically remote part of the world where there is limited data for understanding the climate and hydrology. However the structure of the paper is not easy and clear and the results are not clearly discussed. There are also many type errors. I see many spacing errors between words. On the abstract section alone I have seen more than 10 errors. I have indicated those errors as minor comments. Those errors are too many to list them in my review; I hope the authors will spend some time to correct those errors. With this and other concerns I have indicated below I cannot recommend to accept the paper for publication.

Response: The authors would like to thank the anonymous reviewer for his/her detailed review of our manuscript. We have responded to all comments as given below. We would like to apologies for the spaces between the words, which actually happened during transferring the file to Pdf.

**Specific comments:**

**Abstract**

1. The authors indicated that they have evaluated the performance of the products at various spatial and temporal scale. However, in the abstract Line 26 to 28, the spatial and temporal scale of the evaluation is not indicated.

Response: We have added a description of spatial (point, sub-basin, basin) and temporal (daily, monthly and seasonal) scales to the manuscript in p1 of L28-30.

2. The abstract should be rewritten to summarize the evaluation result at the multiple temporal and spatial scale. The authors indicated that they have done evaluated the 8

products at various temporal (daily, monthly, seasonal) and spatial (point, sub-basin, basin) scales. However, they did not indicated clearly which products worked at what scale.

Response: We have now modified the abstract to show details of the performance, at the different spatial and temporal scales in paragraph 2 and 3.

**Introduction**

3. Page 2: The statement from 23 to 28 needs a reference.

Response: We have included references (Jiang *et al.*, 2012, Guo & Liu, 2016; Dinku *et al.*, 2007; Haile *et al.*, 2010; Dembele *et al.*, 2016) p 3 Line 2 to 3.

4. The authors indicated that tomography as a key factor influencing microclimates in the basin (Page 3 line 32 -33). However, Figure 2b which indicate the relationship between elevation and annual average rainfall doesn't capture the effect of topography. That relationship between rainfall and topography as indicated in this figure is insignificant. What are the author's claming that the topography is a key factor?

Response: We agree with the reviewer that Fig.2b does not show a clear pattern of annual rainfall against the elevation of the station. This contrasts with other basins, where the literature shows that rainfall increases with elevation due to the orographic uplifts (Moreno et al., 2014; Worqlul et al., 2014). However, this pattern is not uniform in the T-A Basin. The annual rainfall increases with elevation in the southern and southwestern parts, while it reduces with elevation in most other parts of the basin. This is shown in Fig. 2a and Fig, 2b: the annual rainfall increases with elevation for some stations and decreases for other stations. This non-uniform pattern is attributed to the complex local topography of the Tekezze-Atbara basin, and seasonal movements of the ITCZ, which influence the microclimate of this basin significantly (Kiros et al., 2016). When the rain-bearing winds reach the basin, their direction is modified by the local topography forcing the release of moisture in the lower areas before they reach the top of mountains. This creates more intense and shorter duration convective rainfall events in the lowlands where warm and moist airflows encounter the mountain foothill (Van der Ent et al., 2010). This shows that topography in the basin plays an important role in moisture cycling either by blocking or capturing moving air masses.

**5. Figure 2b disproves the stament on page 4 line 18 and 19.**

Response: see explanation of point 4 above.

6. The rainfall products were not described very well. As the authors indicated satellite rainfall products quality can be affected by the algorithms used. The authors should discuss the different algorithms and platforms used by those products. What part of the electromagnetic spectrum was used? Are they polar orbiting, or sun synchronized satellites

or a geostationary satellites are used? The description of the different products on page 6 and 7 should address this.

Response: Although these products have been widely applied and documented in the literature, we have improved the description of satellite products as the space may allow in section 2.2.2. More references for detailed information are also included under each product.

7. Page 6 line 21: the autors describe CMORPH product as having a very high spatial and temporal resolution however in the summary table (Table 1) this product doesn't prove to be at a higher resolution compared to others such as CHIRP and ARC and others?

Response: This has been rectified to "The CMORPH product produces global rainfall is also a product from NOAA-CPC"

8. Page 7 line 4. TRMM 3B42V7 is not a latest version

Response: Corrected to "The TRMM product (3B42V7) was developed by the National Aeronautics and Space Administration (NASA)".

9. Table 1 should indicate that the temporal resolution for TRMM3B42V7 should be 3hr. And the product TRMM3B42 should be referred as TMPA-3B42 (Huffman et al., 2010; Prakash et al., 2013; Vrieling et al., 2010).

Response: The temporal resolution is corrected to 3 hourly and the name TRMM 3B42V7 changed to TMPA-3B42 in the table.

10. Page 8 line 22 to 24: Why the inverse distance interpolation is selected? And what was the grid size used for interpolation this will matter since your rainfall products have a various spatial resolution? Inverse distance weighting (IDW) is a possible simple way to go but probably not the best one. There are interpolation algorithms that take into account secondary information (e.g. kriging with external drift).

Response: IDW was adopted in this study for its simple and robust technique which has been commonly applied for rainfall interpolation (e.g. Haile et al., 2010; Jiang et al., 2012; Hu et al., 2014; Worqlul et al., 2014). We agree that each techniques (e.g., IWD and Kriging) have their own advantages and disadvantages; however, the authors believe that applying kriging instead of IWD will not significantly change the results. Further clarification is added to the manuscript in p 10 Line 23 to 26.

12. The performance indicators for satellite rainfall are too simplistic. The authors should consider a categorical statistics to evaluate the effectiveness of those satellite images. Refer Haile et al. (2010). Haile, A.T., Rientjes, T., Gieske, A., Gebremichael, M., 2010. Multispectral remote sensing for rainfall detection and estimation at the source of the Blue Nile River. International Journal of Applied Earth Observation and Geoinformation, 12: S76-S82. The authors should indicate the number of incorrect and correct rain detection by

those satellite products. Why the authors include RMSE and AME is not RMSE better explanatory than AME since it gives higher weight for larger errors. Otherwise, they provide similar outputs.

Response: We agree with the reviewer that the literature shows several alternative statistical indices for performance evaluation of satellite rainfall other than RMSE and MAE. We however maintain that the most commonly used indices include Root Mean Square Error (RMSE), Percent of Bias (PBIAS), Mean Absolut Error (MAE), Mean Error (ME), and correlation coefficient (r). ThEse are used by, e.g., Meng et al. (2013), Dinku et al. (2007), Derin and Yilmaz (2014), Katsnaos et al. (2015), Jiang et al. (2012), Worqlul et al. (2014), among many others.

We agree with the reviewer that these statistical indices are simple but they are the most commonly applied and well documented in the literature as shown above.

We agree that both MAE and RMSE are used to evaluate the average magnitude of the error but RMSE gives a relatively high weight to large errors compared to MAE, implying that RMSE is more useful when large errors are particularly undesirable. The MAE is suitable to describe uniformly distributed errors while the RMSE is more appropriate if the errors are normally distributed (Chai and Draxler, 2014). Thus, evaluating the satellite products using both indices is advantageous. A more detailed explanation is added to the text document, in p 10 Line 31 to 34.

13. The reference use on page 9 line 3 Moriasi et al., 2007 is actually for a performance evaluation of simulated flow, sediment and nutrient. My question is that if you accept a PBIAS of  $\pm$  25 and R of 0.5 (which will be 0.25 R-square) as input to your hydrological model; imagine the performance of your model. I really do not agree with the performance evaluation criteria.

Response: We agree that such criteria are used to evaluate simulated streamflow, and definitely a rainfall PBIAS of 25% will lead to large uncertainty of runoff computed by hydrological models. However, we maintain that still the satellite rainfall data can be useful for applications other than accurate hydrological modelling. We have amended the text in Section 3.2.

**Result**

14. The authors provided a single average statistics like average PBIAS, r, RMSE and MAE for different satellite products (Page 9 line 14 and 15). The authors should discuss the range of variability of those statistics and their relation to landscape position.

Response: We agree that the range of variability is more explanatory than an average value. Although the value of each station is provided as supplementary file (S2-S5) and in Fig.3, we have added the range of statistical indices to the manuscript in Table 3 which is discussed in section 4.1. We have added the standard deviation to the tables. Their relation to the landscape is now discussed in section 4.1 of second paragraph (p11) and second paragraph in P13.

15. The discussion in line 16 page 9 is lamped. The authors should address the range of variation, standard deviation and their relation to landscape postion. Otherwise this doesn't make any sense "Similarly, r value of these products was $\geq$  0.5 in the majority of stations with an average value of 0.52, 0.50 and 0.50, respectively.." What does the average line representing in Figure 3 a and b? what does that implies?

Response: See point 14 above

16. The authors indicated that RMSE and MAE has showed the same trend as PBIAS and r (page 9 line 18 and 19). How is this measured?

Response: The value of RMSE and MAE also shows lower value for these products which implies obtained errors by comparing the satellite rainfall against ground measurement is smaller for CHIRPS, RFEv2 and TRMM products compared to the remaining products. The manuscript is modified accordingly.

17. This doesn't make any sense, the study is about comparing of those products with gauged data, but here they averaged the performance statistics. I guess the authors should discuss the range of performance/variability in terms of spatial and temporal scale for each products since this was indicated on the abstract section as a method (page 1 line 23 and 24).

Response: See point 14 above. We have amended the manuscript to give the range of indicators, and standard deviations. We have also discussed such variability, in section 4.1.

19. Table 2: On Figure 3a I can see a PBIAS value of negative but under Table 2 the autors indicated rage of PBIAS from 0 to infinity. How do you council that?

Response: We have corrected the range value in the Table 2

20. Table 2: The authors should remember that R = 1 doesn't mean perfect, it is obvious we have to check the slope and interest of the fitted line. Eg. Y = 5x + 8 has Pearson correlation coefficient (r) of 1 but Y and X are not similar.

Response: We agree with the reviewer that it may not necessarily be perfect for the given reason. We have now removed such an explanation from the table to avoid confusion.

21. Table 3 is duplicated on Table 4. Remove Table 3

Response: They are not the same, Table 3 is the summary of average (now modified to range value) accuracy indicators from pixel-to-point monthly comparison while Table 4 shows the summary of statistical indices from aerial averaged rainfall comparisons at basin level.

**Figures**

**22. Figure 1. Label the two figures. What does the dotted line over the DEM represent?**

Response: The name of the study area was missed from the legend. We have now improved the figure to include the label. The two dotted lines in the figure represent groups of rainfall stations in the highlands (>2500 m.a.s,l) and lowlands (<2500 m.a.s.l), which is later used to compare the performance of satellite rainfall products in highlands against lowlands. The figure caption is improved to explain this.

**23. Figure 2. Label the two figures and describe them independently.**

Response: Figures are now labelled independently and description of each figure is given under the figure.

**24. Figure 3. What does the average line representing?**

Response: The average line indicated represents the average value of PBIAS and r of all products and is helpful to identify how far the value of each product deviates from the average value of all products. This is also included in the manuscript

**25. Figure 4 where are those representative station located in the watershed?**

Response: Indicating these stations will help readers appreciate the effect of landscapes on the performance of the satellite rainfall. The names of these stations are included in the map (Fig.1)

**General comments:**

**Abstract: the abstract full or problem**

Response: We agree with the reviewer that there were many space errors throughout the document. Space errors were created when the sources document was converted into the Pdf file during uploading. We will cross-check any possible spacing error when uploading the revised manuscript.

**1. Line 21: space between rainfallproducts**

**Response: corrected**

2. Line 26 space between that CHIRPS, Line 26 space between TRMM, and, Line 26 space between wereable

**Response: corrected in the file**

2. Line 27 space between BIAS and

**Response: corrected**

3. Line 28 space between >0.5 over different

**Response: corrected**

4. Line 35 space between respectively.CMORPH

Response: corrected

5. Line 35 space between scale. Their

**Response: corrected**

6. Line 39 space between lowlandswhereas

**Response: corrected**

7. Line 40 space between athighland

**Response: corrected**

8. Line 41 space between the pixel-to-point comparison

**Response: corrected**

9. Line 42 space between showthat

**Response: corrected**

10. Line 42 space between scalesin

**Response: corrected**

11. Page 3 line 29 and 30 modify it as: with a significant elevation variation

**Response: Sentences modified accordingly**

12. Page 6 Line 17: sofar PM and IR are not defined. I see later in the paper they are defined.

Response: They are now defined in the first sentences and abbreviations are used in the remaining document

14. Many many errors (dailyrainfall page 9 line 10, (r)of page 9 line 11, and Tables page 9

line 14, double fullstops (page 9 line 17), MAE, which line 18,

Response: Thank you very much. Such errors were clearly seen in the Pdf file, uploaded in the HESSD website. We will take care of such problems when uploading our revised manuscript.

15. Page 9: wasfurther (line 24), investigatedat, that the, correlationfor, reducedat, Forexample

Response: Space problem is now corrected

16. Page 10: madefor, correlationcoefficients, of the, season.CHIRPS,

Response: Space problem is now corrected

17. Page 11: many

Response: All space problems corrected in the document 18. Page 12: so many type errors

Response: All space problems corrected in the document

19. Page 14: ofTRMM, productshave, withsimilar, werefound, products.Bayissaet, (2017)revealed, (2007)showedthat CMORPH, etcccc

Response: We would like to thank you and all problems corrected in the document

**Reference**

- Haile, A.T., Rientjes, T., Gieske, A., Gebremichael, M., 2010. Multispectral remote sensing for rainfall detection and estimation at the source of the Blue Nile River. International Journal of Applied Earth Observation and Geoinformation, 12: S76-S82.
- Huffman, G.J., Adler, R.F., Bolvin, D.T., Nelkin, E.J., 2010. The TRMM multi-satellite precipitation analysis (TMPA), Satellite rainfall applications for surface hydrology. Springer, pp. 3-22.
- Prakash, S., Mahesh, C., Gairola, R., 2013. Comparison of TRMM Multi-satellite Precipitation Analysis (TMPA)-3B43 version 6 and 7 products with rain gauge data from ocean buoys. Remote sensing letters, 4(7): 677-685.
- Vrieling, A., Sterk, G., de Jong, S.M., 2010. Satellite-based estimation of rainfall erosivity for Africa. Journal of hydrology, 395(3): 235-241.

Response: Thank you for the suggestions we have used these references in our document.

**References**

Asadullah, A., McIntyre, N., & Kigobe, M.: Evaluation of five satellite products for estimation of rainfall over Uganda. *Hydrolog. Sci. J.*, 53(6), 1137-1150, 2008.

- Chai. T and Draxler. R. R.: Root mean square error (RMSE) or mean absolute error (MAE)? Arguments against avoiding RMSE in the literatur. Geosci. Model Dev., 7, 1247–1250, 2014.
- Dembélé, M., & Zwart, S. J.: Evaluation and comparison of satellite-based rainfall products in Burkina Faso, West Africa. *Int. J. Remote Sens.*, *37*(17), 3995-4014, 2016.
- Derin, Y., & Yilmaz, K. K.: Evaluation of multiple satellite-based precipitation products over complex topography. *J. Hydrometeorol.*, *15*(4), 1498-1516, 2014.
- Dinku, T., Ceccato, P., Grover-Kopec, E., Lemma, M., Connor, S., & Ropelewski, C.: Validation of satellite rainfall products over East Africa's complex topography. Int.J.Remote Sens., 28(7), 1503-1526, 2007
- Guo, R., & Liu, Y.: Evaluation of Satellite Precipitation Products with Rain Gauge Data at Different Scales: Implications for Hydrological Applications. *Water*, *8*(7), 28, 2014.
- Hu, Q, Yang D, Li Z, Mishra A, Wang Y, & Yang H.: Multi-scale evaluation of six high-resolution satellite monthly rainfall estimates over a humid region in China with dense rain gauges. *Int. J. Remote Sens.*, 35(4), 2014.
- Huffman, G.J., Adler, R.F., Bolvin, D.T., Nelkin, E.J., 2010. The TRMM multi-satellite precipitation analysis (TMPA), Satellite rainfall applications for surface hydrology. Springer, pp. 3-22.
- Jiang, S., Ren, L., Hong, Y., Yong, B., Yang, X., Yuan, F., & Ma, M.: Comprehensive evaluation of multi-satellite precipitation products with a dense rain gauge network and optimally merging their simulated hydrological flows using the Bayesian model averaging method. J. Hydrol., 452, 213-225, 2012.
- Katsanos, D., Retalis, A., & Michaelides, S.: Validation of a high-resolution precipitation database (CHIRPS) over Cyprus for a 30-year period. *Atmos. Res.*, 169, 459-464, 2015.
- Kiros, G., Shetty, A., & Nandagiri, L.: Analysis of variability and trends in rainfall over northern Ethiopia. *Arab. J.Geosci.*, 9(6), 451, 2016.
- Meng, J., Li, L., Hao, Z., Wang, J., & Shao, Q.: Suitability of TRMM satellite rainfall in driving a distributed hydrological model in the source region of Yellow River. J. Hydrol. 509, 320-332, 2014.
- Moreno, J. F., Mannaerts, C. M., & Jetten, V.: Influence of topography on rainfall variability in Santiago Island, Cape Verde. *Int. J. Climatol.*, 34(4), 1081-1097, 2014.
- Shukla, S., McNally, A., Husak, G., Funk, C.: A seasonal agricultural drought forecast system for foodinsecure regions of East Africa. Hydrol. Earth Syst. Sci. 18, 3907–3921, 2014.
- Thiemig, V., Rojas, R., Zambrano-Bigiarini, M., Levizzani, V., & De Roo, A.: Validation of satellitebased precipitation products over sparsely gauged African river basins. J. Hydrometeoro., 13(6), 1760-1783, 2012.
- Van der Ent, R. J., Savenije, H. H., Schaefli, B., & Steele-Dunne, S. C.: Origin and fate of atmospheric moisture over continents. *Water Resour. Re.*, 46(9), W09525, 2010.

Xu R., F. Tian, L. Yang, H. Hu, H. Lu, and A. Hou.: Ground validation of GPM IMERG and TRMM 3B42V7 rainfall products over southern Tibetan Plateau based on a high-density rain gauge network, J. Geophys. Res. Atmos., 122, 2017